# Gram-weighted Mahalanobis Fréchet Mean: A Hyperparameter-Robust Solution for Model Merging

## Abstract

Model merging has emerged as a promising technique for integrating multiple fine-tuned models into a single unified model without additional training. This paradigm is particularly appealing in resource-constrained scenarios where access to data or retraining is limited. Existing techniques—such as Task Arithmetic, Ties-Merging, and AdaMerging—achieve competitive results but typically rely on extensive hyperparameter tuning, which can be prohibitively expensive for large-scale models. In this work, we propose a hyperparameter-robust merging method that reframes the problem as the estimation of a unified task vector that captures the principal directions of each task (i.e., dominant singular vectors). We formalize this process as the Gram-weighted Mahalanobis Fréchet mean (GMF-Mean), a convex optimization problem that admits a closed-form solution. Our theoretical analysis shows that GMF-Mean inherently adapts to both orthogonal (non-interfering) and conflicting (collinear but opposing) task interactions by automatically modulating the magnitudes of the principal directions. This property alleviates the need for costly hyperparameter tuning that is commonly required in Task Arithmetic-based methods. Empirical results on vision, language, and vision-language models show that GMF Mean achieves competitive performance compared to state-of-the-art baselines, while maintaining the advantages of being training-free, data-free, and hyperparameter-robust. These properties position GMF-Mean as a robust solution for real-world deployment.

## 1 Introduction

Large-scale pre-trained models have become the cornerstone of modern machine learning, enabling generalizable representations that transfer effectively to a wide range of downstream tasks (Zhao et al., 2023). Leveraging massive datasets and expressive architectures, these models, when fine-tuned, have set new performance standards across vision (Dosovitskiy et al., 2021), language (Yang et al., 2025), and multimodal (Radford et al., 2021) domains. This pre-training and fine-tuning paradigm has fueled research into *model merging* (Li et al., 2023)—the integration of multiple fine-tuned models into a single, unified multi-task model. By sidestepping the need for costly retraining, model merging offers a highly attractive solution, particularly when access to data is limited or computational resources are constrained.

A central concept in model merging, introduced by Ilharco et al. (2023), is the *task vector*—the difference between fine-tuned and pre-trained parameters. By adding scaled task vectors to the pre-trained model, task-specific knowledge can be effectively integrated. Building upon this foundation, subsequent methods have refined the merging process through test-time adaptation (Yang et al., 2024b;a) and training-free approaches (Yadav et al., 2023; Sun et al., 2025a).

Despite recent advances, a persistent challenge remains: most existing methods rely on computationally intensive validation to tune hyperparameters for optimal performance (Table 1). Among these, the scaling coefficient applied to task vectors is particularly sensitive—insufficient scaling limits the incorporation of task-specific knowledge, while excessive scaling introduces harmful interference. As a result, extensive hyperparameter tuning is typically required to reach competitive performance.

Table 1: Comparison of resource requirements for representative model merging methods.

| Method | Training Requirement | Data Requirement | Hyperparameter Tuning | Performance |
|---|---|---|---|---|
| Parameter Averaging | None | None | None | Less Competitive |
| Task Arithmetic | None | Required for Tuning | 1 (Scaling coefficient. Sensitive) | Less Competitive |
| AdaMerging | Required at Test-Time | Required for Tuning and Training | Many (LR, batch size, epochs, etc. Sensitive) | Competitive |
| CAT Merging | None | Required for Tuning and Merging | 2 (Scaling coefficient, threshold. Sensitive) | Competitive |
| TSV | None | None | 1 (Scaling coefficient. Not robust in difficult cases) | Competitive |
| GMF Mean (Ours) | None | None | 1 (Scaling coefficient, Robust) | Competitive |

This tuning process is not only computationally expensive, often requiring numerous validation passes, but also practically challenging. Constructing a comprehensive validation set that captures the full spectrum of task capabilities is itself a nontrivial endeavor, especially for large language models (LLMs), whose abilities are vast and not yet fully understood (Chang et al., 2024). These computational and data demands significantly hinder the practical adoption of model merging, especially in the very resource-constrained scenarios where model merging offers the most promise.

In this work, we address this bottleneck by proposing a *hyperparameter-robust* solution for model merging. Our central idea is to compute a merged task vector that adaptively reflects the principal components of the individual task vectors. We formalize this as the *Gram-weighted Mahalanobis Fréchet Mean (GMF Mean)* problem, which seeks to minimize the sum of Mahalanobis distances between the merged vector and task vectors, each weighted by its corresponding Gram matrix. This formulation yields a convex optimization problem with a closed-form solution. Our theoretical analysis reveals that GMF Mean automatically scales task vectors based on their degree of alignment or conflict: when task vectors are orthogonal, GMF Mean reduces to their simple sum; when they are collinear and conflicting, it becomes a singular-value–weighted average. This adaptive scaling obviates the need for manual tuning and ensures that GMF Mean retains the advantages of being training-free, data-free, and hyperparameter-robust.

In summary, the contributions of this paper are as follows:

- We introduce a novel perspective on model merging by reformulating it as a Gram-weighted Mahalanobis Fréchet Mean (GMF Mean) problem, yielding a closed-form, hyperparameter-robust solution.

- We show that GMF Mean performs conflict-adaptive scaling: it reduces to simple summation for orthogonal task vectors and to a singular-value–weighted average in the collinear/conflicting directions, eliminating the need for manual scaling.

- We empirically validate GMF Mean across vision, language, and vision-language models, showing that it achieves competitive performance with state-of-the-art baselines, while remaining training-free, data-free, and hyperparameter-robust.

## 2 RELATED WORK

Model merging has emerged as a promising paradigm for integrating multiple fine-tuned models into a single unified model, which is appealing where data is inaccessible or computational resources are constrained. One of the most influential baselines is Task Arithmetic (Ilharco et al., 2023), which introduces the concept of task vectors—defined as the difference between the parameters of a fine-tuned model and its pre-trained initialization. Task Arithmetic shows that adding these vectors in parameter space can accumulate knowledge from different tasks. While simple and effective, Task Arithmetic often struggles when task vectors exhibit conflicts, leading to sub-optimal performance.

To enhance Task Arithmetic, several works have explored test-time adaptation techniques. These methods update model parameters or merging weights on-the-fly using task-specific signals extracted from unlabeled test data. For example, AdaMerging (Yang et al., 2024b) adjusts per-task weights through entropy-based objectives; Surgery (Yang et al., 2024a) and Prodistill (Xu et al., 2025) align intermediate representations; WEMoE (Tang et al., 2024) and Twin Merging (Lu et al., 2024) mixture-of-experts frameworks with learned routers. Despite their effectiveness, such methods require access to data at inference and involve costly iterative updates, limiting their practicality.

Recent research has increasingly focused on training-free model merging. Some methods, such as RegMean (Jin et al., 2023) and CAT Merging (Sun et al., 2025a), rely on representative exem-

plars and utilize their feature distributions to facilitate harmonious merging. Other approaches, such as Ties-Merging (Yadav et al., 2023) and PCB Merging (DU et al., 2024), eliminate the need for representative exemplars by focusing on the large-magnitude components of task vectors. Recent Singular Value Decomposition (SVD)-based methods, including TSV (Gargiulo et al., 2025) and Iso-Merging (Marczak et al., 2025), emphasize retaining the principal components of task vectors. These methods demonstrate robustness to hyperparameter selection, thus alleviating the need for manual tuning, though their robustness degrades when scaling to many models or complex multi-modal architectures. In contrast, the GMF Mean proposed in this paper is training-free, data-free, and exhibits superior robustness to hyperparameter selection across a wide range of scenarios. Notably, techniques such as Ties-Merging (Yadav et al., 2023) and Localize-and-Stitch (He et al., 2025) provide recommended default hyperparameter values; however, their performance remains sensitive to hyperparameter choices, often showing substantial improvement with additional validation.

## 3 PRELIMINARY

### 3.1 MODEL MERGING

**Problem setup.** Consider a pre-trained model parameterized by $W_{\text{pre}} = \{W_{\text{pre}}^{(l)}\}_{l=1}^{L}$, where $W_{\text{pre}}^{(l)} \in \mathbb{R}^{d_{\text{in}} \times d_{\text{out}}}$ denotes the parameters of the $l$-th layer, and $d_{\text{in}}$, $d_{\text{out}}$ are its input and output feature dimensions. After fine-tuning on $K$ downstream tasks, we obtain task-specific models $\{W_1, \dots, W_K\}$. The goal of model merging is to construct a merged model $W_{\text{mtl}}$ that performs well across all $K$ tasks without requiring costly retraining.

**Baseline: Task Arithmetic (Ilharco et al., 2023).** Task Arithmetic introduces the *task vector*, defined as the parameter difference between fine-tuned expert $W_k$ and the pre-trained model $W_{\text{pre}}$:

$$T_k = W_k - W_{\text{pre}}, \tag{1}$$

where the subtraction is layer-wise: $W_k - W_{\text{pre}} = \{W_k^{(l)} - W_{\text{pre}}^{(l)}\}_{l=1}^{L}$. Task Arithmetic shows that adding scaled task vectors to the pre-trained model can effectively integrate task knowledge:

$$W_{\text{mtl}}^{\text{ta}} = W_{\text{pre}} + \lambda \sum_{k=1}^{K} T_k, \tag{2}$$

where $\lambda$ is a manually chosen scaling factor.

Equivalently, Task Arithmetic computes the Euclidean barycenter of the task vectors for each layer:

$$\bar{T}^{(l)} = \arg\min_{T^{(l)}} \sum_{k=1}^{K} \|T^{(l)} - T_k^{(l)}\|_2^2 = \frac{1}{K} \sum_{k=1}^{K} T_k^{(l)}, \quad W_{\text{mtl}}^{\text{ta},(l)} = W_{\text{pre}}^{(l)} + \lambda \bar{T}^{(l)}. \tag{3}$$

However, when task vectors are in conflict—for example, when they point in opposite directions or exhibit strong anisotropy—this simple estimator becomes inadequate, and the performance becomes highly sensitive to the manually chosen hyperparameter $\lambda$ (Sun et al., 2025a). To address this, we adopt a task-aware aggregation by replacing the Euclidean metric with the Mahalanobis metric, as described in the following sections. For clarity, we focus on layer-wise merging and omit the layer index, treating $T_k$ as the task vector for a given layer.

### 3.2 MAHALANOBIS DISTANCE

**Definition 3.1** (Mahalanobis distance (Mahalanobis, 1930)). Given a set $\mathcal{X}$ and two points $x_i, x_j \in \mathcal{X}$, the Mahalanobis distance between $x_i$ and $x_j$ is defined as

$$\|x_i - x_j\|_{\Sigma^{-1}} := \sqrt{(x_i - x_j)^\top \Sigma^{-1} (x_i - x_j)}, \tag{4}$$

where $\Sigma$ is the covariance matrix of $\mathcal{X}$.

**Definition 3.2** (Generalized Mahalanobis distance (Ghojogh et al., 2022)). Replacing $\Sigma^{-1}$ with a positive semi-definite weight matrix $C \succeq 0$ generalizes the metric:

$$\|x_i - x_j\|_C := \sqrt{(x_i - x_j)^\top C (x_i - x_j)}. \tag{5}$$

For matrix-valued inputs $X_i, X_j \in \mathbb{R}^{m \times n}$, with $\Delta = X_i - X_j$, a trace-based extension with a row-weight matrix $C \succeq 0$ ($C \in \mathbb{R}^{m \times m}$) is

$$\|X_i - X_j\|_C := \sqrt{\operatorname{tr}\left(\Delta^\top C \Delta\right)} = \left\|C^{1/2}\Delta\right\|_F. \tag{6}$$

Intuitively, the Mahalanobis distance corresponds to a Euclidean distance under whitening or anisotropic scaling. By choosing $C$, one can emphasize specific directions: if $C = U \operatorname{diag}(\lambda_1, \ldots, \lambda_d)U^\top$ with $\lambda_i \geq 0$, then directions aligned with $U_i$ with larger $\lambda_i$ contribute more to the distance. This property is especially useful when the principal components of $C$ capture the most task-relevant information.

### 3.3 FRÉCHET MEAN

**Definition 3.3** (Fréchet Mean (Fréchet, 1948))**.** Let $(\mathcal{X}, d)$ be a metric space with distance function $d : \mathcal{X} \times \mathcal{X} \to [0, \infty)$. For points $\{x_k\}_{k=1}^K \subset \mathcal{X}$, the Fréchet mean is any minimizer:

$$\mu^\star \in \arg\min_{x \in \mathcal{X}} \Phi(x), \quad \text{where} \quad \Phi(x) = \sum_{k=1}^K d^2(x, x_k). \tag{7}$$

If $\Phi(x)$ is strictly convex, the mean is unique.

In model merging, the points $x_k$ correspond to task vectors or parameter vectors. The Fréchet mean provides an aggregation under a specified metric $d$, which shapes the characteristics of the solution.

## 4 GMF MEAN FOR MODEL MERGING

Recent advances in model merging have revealed that, despite their high dimensionality, task vectors often reside in low-dimensional subspaces (Gargiulo et al., 2025). In particular, their principal components capture the most informative directions for generalization across tasks. Motivated by this observation, our key insight is to construct a merged task vector that aligns with these principal components—the directions that explain the most significant variance among task vectors—where such alignment can be naturally quantified using the Mahalanobis distance.

Specifically, we reformulate the model merging objective as the optimization of the *Gram-weighted Mahalanobis Fréchet Mean* (GMF Mean). For each layer, we seek a merged task vector $T^\star$ that solves:

$$T^\star \in \arg\min_T \sum_{k=1}^K \|T - T_k\|_{T_k T_k^\top}^2, \tag{8}$$

where $\|T - T_k\|_{T_k T_k^\top}$ denotes the generalized Mahalanobis distance weighted by the Gram matrix $T_k T_k^\top$. We select $T_k T_k^\top$ as the weight since it naturally emphasizes the principal directions of $T_k$. If $T_k$ is represented via singular value decomposition as $T_k = U \operatorname{diag}(\sigma_1, \ldots, \sigma_d)V^\top$, then $T_k T_k^\top = U \operatorname{diag}(\sigma_1^2, \ldots, \sigma_d^2)U^\top$, which assigns greater importance to directions with larger singular values (see Section 5 for further discussion). Additionally, $T_k T_k^\top$ is positive semi-definite, ensuring the convexity of equation 8. It is worth noting that equation 8 does not strictly adhere to the classical definition of the Fréchet mean, as the distance metric varies across data points (i.e., task vectors). We refer to equation 8 as a *task-adaptive generalization* of the Fréchet mean, which preserves convexity while enabling the aggregation to account for task-specific information.

This objective admits a closed-form solution (see Section A.1 for proof):

$$T^\star = \left(\sum_k T_k T_k^\top\right)^\dagger \left(\sum_k T_k T_k^\top T_k\right), \tag{9}$$

where $\dagger$ denotes the Moore-Penrose pseudo-inverse. This yields the minimum-norm solution for the merged task vector. For one-dimensional vectors (e.g., bias terms), $T_k \in \mathbb{R}^d$ can be viewed as a degenerate case of a $1 \times d$ matrix.

As summarized in Algorithm 1, the merged model is obtained by adding $T^\star$ into $W_{\text{pre}}$:

$$W_{\text{mtl}}^{\text{gmf}} = W_{\text{pre}} + \lambda T^\star. \tag{10}$$

From extensive empirical evidence (Section 6.4 and C.4), we observe that $T^\star$ exhibits strong robustness to scaling and consistently achieves competitive performance around $\lambda = 1.0$. In Section A.7, we further empirically validate that this hyperparameter-robust nature is stable to variations in the relative orientation or scale of the Gram matrices, as long as their magnitudes remain within a comparable range—an easily satisfied condition in typical model-merging scenarios. Such hyperparameter robustness substantially reduces tuning overhead in practical deployments. In the next section, we will demonstrate that GMF Mean adaptively reweights task vectors by their conflict magnitude, yielding robust and practical model merging.

---

**Algorithm 1:** GMF Mean Model Merging

---

**Input:** Pre-trained model $W_{\text{pre}}$; Task vectors $\{T_1, \ldots, T_K\}$

**Output:** Merged model $W_{\text{mtl}}^{\text{gmf}}$

1 **for** $l = 1$ **to** $L$ **do**

2  $\quad T^{(l)^\star} = \left( \sum_k T_k^{(l)} T_k^{(l)^\top} \right)^\dagger \left( \sum_k T_k^{(l)} T_k^{(l)^\top} T_k^{(l)} \right);$      // Refer to equation 9

3 $T^\star = \{T^{(1)^\star}, \ldots, T^{(L)^\star}\}$

4 $W_{\text{mtl}}^{\text{gmf}} = W_{\text{pre}} + \lambda T^\star$

5 **return** $W_{\text{mtl}}^{\text{gmf}}$

---

## 5    DISCUSSION: THE ADAPTABILITY OF GMF MEAN

We provide a theoretical analysis to elucidate the adaptability of GMF Mean. Our analysis leverages the singular value decomposition of each task vector, $T_k = U_k \Sigma_k V_k^\top$.

**Non-conflicting principal directions (Figure 1a).** Consider the scenario where only a single task vector $T_i$ exhibits significant energy along a pair of principal directions $(u, v)$, with $u \in U_i$ and $v \in V_i$, while for all $k \neq i$, $u^\top U_k = 0$ and $v^\top V_k = 0$. In this setting, $T_i^\top u = \sigma v$ and $T_i v = \sigma u$ for some singular value $\sigma$, while

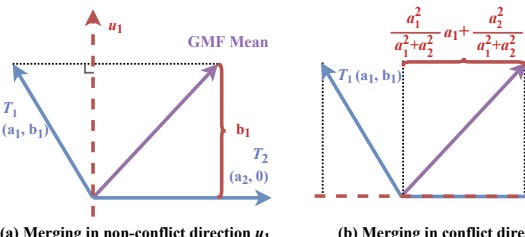

(a) Merging in non-conflict direction $u_1$      (b) Merging in conflict direction $u_2$

Figure 1: Illustration of GMF Mean in non-conflicting (a) and conflicting (b) settings when merging two task vectors $T_1$ and $T_2$.

$T_k^\top u = T_k v = 0$ for $k \neq i$. Under these conditions, it can be shown (see Section A.2) that the GMF Mean solution satisfies:

$$T^{\star\top} u = \sigma v \quad \text{and} \quad T^\star v = \sigma u, \tag{11}$$

which implies that $(u, v)$ remain singular vectors of the merged solution $T^\star$ with singular value $\sigma$. Thus, in non-conflicting cases, GMF Mean preserves the principal components contributed by individual task vectors without distortion.

**Conflicting or shared principal directions (Figure 1b).** In another situation where multiple task vectors share energy along a common pair of singular directions $(u, v)$, the GMF Mean solution becomes an adaptive weighted combination (see Section A.3 for details):

$$T^{\star\top} u = \underbrace{\left( \sum_k \frac{\sigma_k^2}{\sum_k \sigma_k^2} \sigma_k \right)}_{\text{Weighted Average}} v, \qquad T^\star v = \left( \sum_k \frac{\sigma_k^2}{\sum_k \sigma_k^2} \sigma_k \right) u, \tag{12}$$

Table 2: Multi-task performance when merging ViT-B/32 models on eight vision tasks. The best performance among hyperparameter-robust methods is highlighted with **bold**. The "#best" column represents the number of datasets where the hyperparameter-robust method performs the best.

| Method | SUN397 | Cars | RESISC45 | EuroSAT | SVHN | GTSRB | MNIST | DTD | Avg Acc | #best |
|---|---|---|---|---|---|---|---|---|---|---|
| | | | | *Basic baseline methods* | | | | | | |
| Pre-trained | 62.3 | 59.7 | 60.7 | 45.5 | 31.4 | 32.6 | 48.5 | 43.8 | 48.0 | - |
| Individual | 75.3 | 77.7 | 96.1 | 99.7 | 97.5 | 98.7 | 99.7 | 79.4 | 90.5 | - |
| Traditional MTL | 73.9 | 74.4 | 93.9 | 98.2 | 95.8 | 98.9 | 99.5 | 77.9 | 88.9 | - |
| | | | | *Test-time training-based methods* | | | | | | |
| AdaMerging | 64.5 | 68.1 | 79.2 | 93.8 | 87.0 | 91.9 | 97.5 | 59.1 | 80.1 | - |
| AdaMerging++ | 66.6 | 68.3 | 82.2 | 94.2 | 89.6 | 89.0 | 98.3 | 60.6 | 81.1 | - |
| Surgery Merging | 63.8 | 59.9 | 83.3 | 97.9 | 87.0 | 87.0 | 98.6 | 69.4 | 80.9 | - |
| Localize-and-Stitch | 67.2 | 68.3 | 81.8 | 89.4 | 87.9 | 86.6 | 94.8 | 62.9 | 79.9 | - |
| DOGE AM | 70.5 | 74.8 | 88.7 | 94.1 | 91.6 | 95.7 | 98.8 | 72.5 | 85.9 | - |
| | | | | *Training-free methods* | | | | | | |
| Weight Averaging | 65.3 | 63.4 | 71.4 | 71.7 | 64.2 | 52.8 | 87.5 | 50.1 | 65.8 | - |
| Fisher Merging | 68.6 | 69.2 | 70.7 | 66.4 | 72.9 | 51.1 | 87.9 | 59.9 | 68.3 | - |
| RegMean | 65.3 | 63.5 | 75.6 | 78.6 | 78.1 | 67.4 | 93.7 | 52.0 | 71.8 | - |
| Task Arithmetic | 55.2 | 54.9 | 66.7 | 78.9 | 80.2 | 69.7 | 97.3 | 50.4 | 69.1 | - |
| Ties-Merging | 59.8 | 58.6 | 70.7 | 79.7 | 86.2 | 72.1 | 98.3 | 54.2 | 72.4 | - |
| TATR | 62.7 | 59.3 | 72.3 | 82.3 | 80.5 | 72.6 | 97.0 | 55.4 | 72.8 | - |
| Consensus Merging | 65.7 | 63.6 | 76.5 | 77.2 | 81.7 | 70.3 | 97.0 | 57.1 | 73.6 | - |
| AWD Merging | 63.5 | 61.9 | 72.6 | 84.9 | 85.1 | 79.1 | 98.1 | 56.7 | 75.2 | - |
| PCB Merging | 63.8 | 62.0 | 77.1 | 80.6 | 87.5 | 78.5 | 98.7 | 58.4 | 75.8 | - |
| CAT Merging | 68.1 | 65.4 | 80.5 | 89.5 | 85.5 | 78.5 | 98.6 | 60.7 | 78.3 | - |
| DOGE TA | 67.7 | 70.1 | 82.0 | 90.3 | 86.3 | 86.8 | 98.3 | 64.0 | 80.7 | - |
| LOT Merging | 67.7 | 67.5 | 85.7 | 94.9 | 93.4 | 89.8 | 98.7 | 63.6 | 82.7 | - |
| FR-Merging | 66.2 | 64.5 | 77.2 | 90.1 | 85.4 | 82.3 | 98.5 | 60.0 | 78.1 | - |
| ISO-C | 71.7 | 70.5 | 87.0 | 95.0 | 90.0 | 90.8 | 99.2 | 68.6 | 84.1 | - |
| ISO-CTS | 74.5 | 74.1 | 88.9 | 93.6 | 84.7 | 90.1 | 98.8 | 69.8 | 84.3 | - |
| | | | | *Hyperparameter-robust methods* | | | | | | |
| TSV | 70.1 | **72.1** | **85.9** | **94.3** | 90.9 | 91.2 | 99.2 | 68.8 | 84.1 | 3 |
| GMF Mean (ours) | **71.3** | 69.2 | 83.7 | 94.2 | **93.2** | **92.8** | **99.4** | **69.7** | **84.2** | 5 |

where $\sigma_k$ denotes the singular value of $T_k$ in the $(u, v)$ direction. Here, $(u, v)$ remain the singular vectors of $T^\star$, but with a singular value adaptively determined as $\sigma_{\text{GMF}} = \sum_k \frac{\sigma_k^2}{\sum_k \sigma_k^2} \sigma_k$. Each task's contribution is automatically reweighted in proportion to $\sigma_k^2$, so that dominant directions have a greater influence on the merged solution, while weaker directions are downweighted.

The above analysis highlights the self-adaptive mechanism of GMF Mean: it exactly preserves unique principal components of individual tasks, while reconciling shared components through a principled, variance-aware weighting scheme. This adaptivity underlies GMF Mean's robustness to scaling and alleviates the need for manual tuning. For a more general discussion based on global principal directions, we refer the reader to Section A.4 and A.5.

## 6 EXPERIMENTS

### 6.1 SETTINGS

**Benchmarks.** We evaluate GMF Mean across vision, language, and vision-language tasks:

- For the vision tasks, we utilize the commonly used eight image classification datasets: SUN397 (Xiao et al., 2016), Cars (Krause et al., 2013), RESISC45 (Cheng et al., 2017), EuroSAT (Helber et al., 2019), SVHN (Netzer et al., 2011), GTSRB (Stallkamp et al., 2011), MNIST (LeCun & Cortes, 2010), and DTD (Cimpoi et al., 2014);

- For the language tasks, we select four tasks from the MTEB benchmark (Muennighoff et al.): ArguAna (Wachsmuth et al., 2018), HotpotQA (Yang et al., 2018), BIOSSES (Soğancıoğlu et al., 2017), and Banking77 (Casanueva et al., 2020);

- For the vision-language tasks, we focus on three captioning datasets (COCO Caption (Chen et al., 2015), Flickr30k Caption (Plummer et al., 2015), Textcaps (Sidorov et al., 2020)) and three Visual Question Answering (VQA) datasets (OKVQA (Marino et al., 2019), TextVQA (Singh et al., 2019), and ScienceQA (Lu et al., 2022)).

Table 3: Multi-task performance when merging ViT-L/14 models on eight vision tasks.

| Method | SUN397 | Cars | RESISC45 | EuroSAT | SVHN | GTSRB | MNIST | DTD | Avg Acc | #best |
|---|---|---|---|---|---|---|---|---|---|---|
| *Basic baseline methods* | | | | | | | | | | |
| Pre-trained | 66.8 | 77.7 | 71.0 | 59.9 | 58.4 | 50.5 | 76.3 | 55.3 | 64.5 | - |
| Individual | 82.3 | 92.4 | 97.4 | 100.0 | 98.1 | 99.2 | 99.7 | 84.1 | 94.2 | - |
| Traditional MTL | 80.8 | 90.6 | 96.3 | 96.3 | 97.6 | 99.1 | 99.6 | 84.4 | 93.5 | - |
| *Test-time training-based methods* | | | | | | | | | | |
| AdaMerging | 79.0 | 90.3 | 90.8 | 96.2 | 93.4 | 98.0 | 99.0 | 79.9 | 90.8 | - |
| AdaMerging++ | 79.4 | 90.3 | 91.6 | 97.4 | 93.4 | 97.5 | 99.0 | 79.2 | 91.0 | - |
| Surgery Merging | 75.7 | 84.4 | 93.1 | 98.8 | 91.3 | 93.4 | 99.1 | 76.1 | 89.0 | - |
| Localize-and-Stitch | 74.4 | 78.0 | 86.0 | 94.6 | 93.4 | 92.5 | 98.5 | 74.9 | 86.5 | - |
| DOGE AM | 79.7 | 91.6 | 94.4 | 96.7 | 96.5 | 98.6 | 99.0 | 84.1 | 92.6 | - |
| *Training-free methods* | | | | | | | | | | |
| Weight Averaging | 72.1 | 81.6 | 82.6 | 91.9 | 78.2 | 70.7 | 97.1 | 62.8 | 79.6 | - |
| Fisher Merging | 69.2 | 88.6 | 87.5 | 93.5 | 80.6 | 74.8 | 93.3 | 70.0 | 82.2 | - |
| RegMean | 73.3 | 81.8 | 86.1 | 97.0 | 88.0 | 84.2 | 98.5 | 60.8 | 83.7 | - |
| Task Arithmetic | 73.9 | 82.1 | 86.6 | 94.1 | 87.9 | 86.7 | 98.9 | 65.6 | 84.5 | - |
| Ties-Merging | 76.5 | 85.0 | 89.3 | 95.7 | 90.3 | 83.3 | 99.0 | 68.8 | 86.0 | - |
| TATR | 74.6 | 83.7 | 87.6 | 93.7 | 88.6 | 88.1 | 99.0 | 66.8 | 85.3 | - |
| Consensus Merging | 75.0 | 84.3 | 89.4 | 95.6 | 88.3 | 82.4 | 98.9 | 68.0 | 85.2 | - |
| AWD Merging | 76.2 | 85.4 | 88.7 | 96.1 | 92.4 | 92.3 | 99.3 | 69.4 | 87.5 | - |
| PCB Merging | 76.2 | 86.0 | 89.6 | 95.9 | 89.9 | 92.3 | 99.2 | 71.4 | 87.6 | - |
| CAT Merging | 78.7 | 88.5 | 91.1 | 96.3 | 91.3 | 95.7 | 99.4 | 75.7 | 89.6 | - |
| DOGE TA | 76.7 | 87.7 | 91.6 | 96.2 | 94.4 | 93.4 | 98.9 | 71.6 | 88.8 | - |
| LOT Merging | 76.7 | 88.6 | 91.7 | 98.7 | 97.1 | 95.7 | 99.5 | 76.4 | 90.5 | - |
| FR-Merging | 76.4 | 87.0 | 90.2 | 96.8 | 92.0 | 92.8 | 99.3 | 71.5 | 88.3 | - |
| ISO-C | 82.9 | 90.6 | 95.4 | 99.0 | 94.2 | 97.0 | 99.4 | 81.0 | 92.5 | - |
| ISO-CTS | 83.1 | 91.9 | 96.0 | 99.0 | 93.8 | 97.9 | 99.5 | 82.4 | 93.0 | - |
| *Hyperparameter-robust methods* | | | | | | | | | | |
| TSV | 79.2 | 89.8 | 93.8 | **98.9** | 95.6 | 96.5 | 99.5 | 80.0 | 91.7 | 1 |
| GMF Mean (ours) | **82.3** | **90.7** | **95.0** | 98.6 | **96.3** | 97.1 | **99.6** | **81.1** | **92.6** | 7 |

**Baselines.** We compare our approach against a diverse set of methods, categorized into basic baseline methods, test-time training-based model merging methods, training-free model merging methods, and hyperparameter-robust methods. Basic baseline methods provide the performance of the Pre-trained model, the Individual task model, and the Traditional Multi-Task Learning model. For test-time training-based methods, we select AdaMerging, AdaMerging++ (Yang et al., 2024b), Surgery (Yang et al., 2024a), Localize-and-Stitch (He et al., 2025), and DOGE AM (Wei et al., 2025). Among the training-free methods, we report the performance of simple Weight Averaging, Fisher Merging (Matena & Raffel, 2022), RegMean (Jin et al., 2023), Task Arithmetic (Ilharco et al., 2023), Ties-Merging (Yadav et al., 2023), TATR (Sun et al., 2025b), Consensus Merging (Wang et al., 2024a), AWD Merging (Xiong et al., 2024), PCB Merging (DU et al., 2024), CAT Merging (Sun et al., 2025a), DOGE TA (Wei et al., 2025), LOT Merging (Sun et al., 2025c), FR-Merging (Zheng & Wang, 2025), ISO-C, and ISO-CTS (Marczak et al., 2025). Finally, we select TSV (Zhang & Yang, 2022) as a hyperparameter-robust baseline.

**Implementation details.** Our implementation strictly follows the setting in Task Arithmetic (Ilharco et al., 2023). For vision tasks, we use CLIP (Radford et al., 2021) with ViT-B/32 and ViT-L/14 as pre-trained backbones. Task vectors are obtained by fine-tuning on each dataset provided by Ilharco et al. (2023). We report per-task post-merge accuracy and mean accuracy (Avg. Acc.). For language tasks, we adopt Qwen3-0.6B and Qwen3-8B as the pretrained backbones (Yang et al., 2025), and merge two official fine-tuned variants—Qwen3-Embedding and Qwen3-Reranker—with both the 0.6B and 8B versions. For vision-language tasks, task vectors are generated by fine-tuning the VQA version of BLIP Shi et al. (2021) for 6000 steps per task. We fine-tune the VQA variant of BLIP for 6,000 steps per task to obtain task vectors. All parameters (including the image encoder, text encoder, and text decoder) are fine-tuned. Further details are available in the supplementary code.

## 6.2 COMPARING RESULTS

**Performance on merging vision classification models.** Tables 2 and 3 report the results for merging ViT-B/32 and ViT-L/14 models, respectively. As expected, test-time training-based methods generally outperformed training-free methods due to the incorporation of additional training and data. Nevertheless, some advanced training-free methods, such as LOT Merging and ISO-CTS, also

Table 4: Multi-task performance when merging Qwen3 models on four language tasks.

| Method | Model | Ranking-Oriented Tasks | | Embedding-Oriented Tasks | | #best |
| | | ArguAna | HotpotQA | BIOSSES | Banking77 | |
|---|---|---|---|---|---|---|
| **Metric** | | NDCG@10 | NDCG@10 | Pearson Correlation | Accuracy | |
| Task Arithmetic | | 0.55 | 0.50 | 0.83 | 74.67% | 0 |
| LOT Merging | Qwen3-0.6B | 0.60 | 0.49 | **0.85** | 79.40% | 1 |
| TSV | | 0.61 | 0.60 | 0.82 | 75.51% | 0 |
| GMF Mean (ours) | | **0.71** | **0.68** | **0.85** | **80.59%** | 4 |
| Task Arithmetic | | 0.63 | 0.64 | 0.85 | 80.97% | 0 |
| LOT Merging | Qwen3-8B | 0.64 | 0.52 | 0.84 | 85.22% | 0 |
| TSV | | 0.72 | **0.75** | 0.86 | 84.72% | 1 |
| GMF Mean (ours) | | **0.74** | **0.75** | **0.88** | **85.44%** | 4 |

Table 5: Multi-task performance when merging BLIP models on six vision-language tasks.

| Method | COCO Caption | Flickr30k Caption | Textcaps | OKVQA | TextVQA | ScienceQA | #best |
|---|---|---|---|---|---|---|---|
| **Metric** | CIDEr | CIDEr | CIDEr | Accuracy | Accuracy | Accuracy | |
| Pre-trained · | 0.07 | 0.03 | 0.05 | 42.80 | 21.08 | 40.50 | - |
| Individual | 1.17 | 0.65 | 0.65 | 50.84 | 29.79 | 76.89 | - |
| Task Arithmetic | 0.86 | 0.50 | 0.39 | 17.71 | 0.49 | 40.10 | 0 |
| Ties-Merging | 0.53 | 0.27 | 0.22 | 27.95 | 0.57 | 40.35 | 0 |
| TATR | 0.46 | 0.31 | 0.21 | 28.30 | 14.74 | 42.98 | 0 |
| PCB Merging | 0.71 | 0.52 | 0.30 | 36.04 | 1.88 | 43.01 | 0 |
| CAT Merging | 0.91 | 0.53 | 0.36 | **44.07** | 19.69 | 46.36 | 1 |
| LOT Merging | 0.91 | 0.54 | **0.44** | 38.35 | 20.82 | 48.24 | 1 |
| TSV | 0.84 | 0.55 | 0.38 | 30.02 | **24.38** | 52.65 | 1 |
| GMF Mean (ours) | **0.94** | **0.58** | 0.41 | 41.74 | 20.06 | **53.64** | 3 |

demonstrated competitive performance. Our GMF Mean performs the best among hyperparameter-robust methods, demonstrating the highest average accuracy (84.2%) when merging ViT-B/32 models and delivering the best performance on 5 tasks. For the ViT-L/14 models, GMF Mean attains the highest average accuracy (92.6%) and performs the best on 7 tasks compared with TSV. These results highlight the effectiveness of GMF Mean in merging vision models. Although ISO-CTS outperforms GMF Mean in certain merging tasks, as demonstrated in Section C.3, we empirically observe that GMF Mean achieves better performance in many model merging scenarios.

**Performance on merging language models.** Table 4 reports results when merging Qwen3-Embedding and Qwen3-Reranker. Both Qwen3-0.6B and Qwen3-8B were evaluated on ranking-oriented tasks (ArguAna and HotpotQA) and embedding-oriented tasks (BIOSSES and Banking77). For Qwen3-0.6B, LOT Merging achieved the best score on BIOSSES (Pearson correlation 0.85), while TSV performed strongly on ranking tasks. For Qwen3-8B, TSV excelled on HotpotQA, and LOT Merging yielded a notable improvement on Banking77. Across both model scales, however, the proposed GMF Mean method consistently reached the highest performance on all tasks, confirming its robustness for merging language models.

**Performance on merging vision-language models.** The results of merging BLIP models across six vision-language tasks are presented in Table 5. We can see that CAT Merging achieves the best performance on OKVQA with an accuracy of 44.07%, while LOT Merging excels on the Textcaps task with a CIDEr score of 0.44. TSV performs best on TextVQA, achieving the highest accuracy of 24.38%. Despite the strong performance of these methods, the proposed GMF Mean method achieves the best on most tasks, with the best results on COCO Caption (0.94), Flickr30k Caption (0.58), and ScienceQA (53.64%). These findings highlight the effectiveness of GMF Mean in merging vision-language models.

## 6.3 ANALYSIS OF COMPUTATIONAL COMPLEXITY

This section evaluates the computational complexity of GMF Mean. Our GMF Mean is designed for low computational overhead: the merging process is entirely training-free and only relies on a closed-form, layer-wise procedure that involves matrix inversions. While the theoretical complexity of inverting a $d \times d$ matrix is $\mathcal{O}(d^3)$, our implementation uses parallelized GPU kernels and

Table 6: Computational complexity comparison (in seconds) for merging ViT-B/32 and ViT-L/14 models across eight vision tasks, measured on a single RTX 3090 GPU.

| Method | Surgery | AdaMerging | TATR | PCB Merging | CAT Merging | LOT Merging | TSV | GMF Mean (ours) |
|--------|---------|-----------|------|-------------|-------------|-------------|-----|------------------|
| ViT-B/32 | 12621 | 8276 | 176 | 43 | 46 | 44 | 28 | 20 |
| ViT-L/14 | 36826 | 16299 | 283 | 131 | 150 | 161 | 95 | 88 |

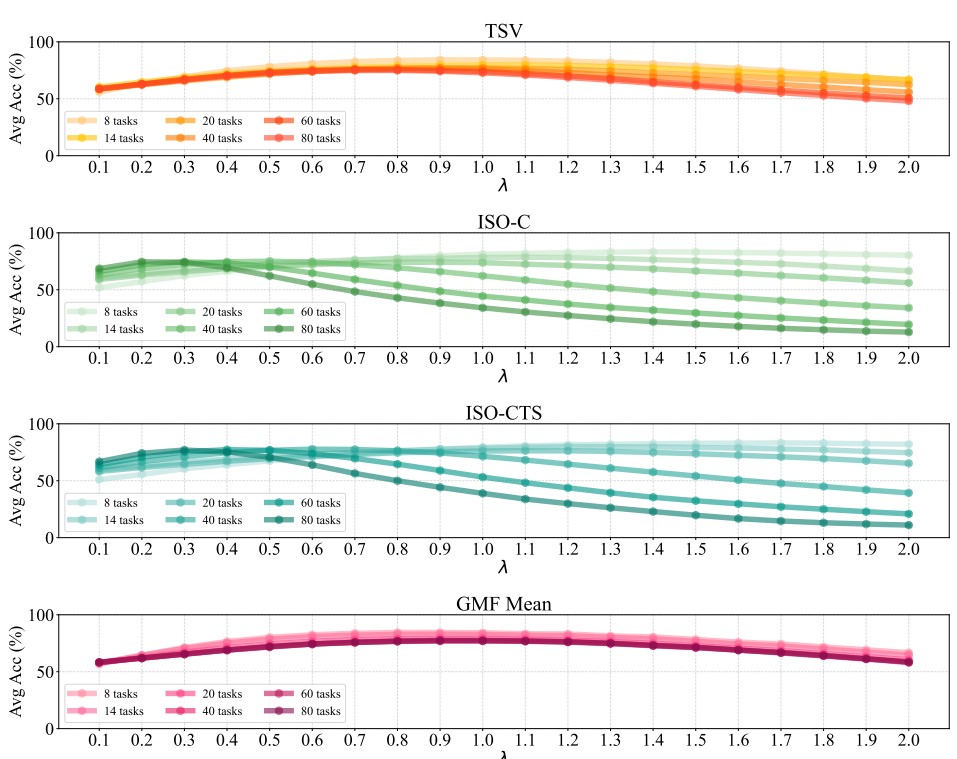

Figure 2: Average accuracy (%) when merging eight ViT-B/32 models versus scaling factor $\lambda$.

batched linear algebra, yielding efficient runtime in practice. As reported in Table 6, GMF Mean achieves substantially lower wall-clock time. Moreover, GMF-Mean exhibits strong robustness to hyperparameter selection, thereby further reducing the tuning overhead in practical applications.

## 6.4 SENSITIVE ANALYSIS OF $\lambda$

In this paper, we introduce the hyperparameter-robust GMF Mean. To verify this benefit, this section conducts a sensitivity analysis by sweeping $\lambda$ and evaluating the average accuracy of all tasks. We assess three competitive baselines—TSV, ISO, and ISO-CTS—on the ViT-B/32 backbone with $\lambda \in \{0.1, 0.2, \ldots, 2.0\}$. We merge 14 and 20 tasks provided by Wang et al. (2024b), and further expand the 20-task scenario to 40, 60, and 80 tasks by modifying the classification prompt templates used to generate label embeddings.[1]

As illustrated in Figure 2, TSV, ISO, and ISO-CTS remain relatively robust around $\lambda = 1.0$ when the task set is small. However, as the number of tasks grows, their optimal $\lambda$ shifts substantially, resulting in fragile hyperparameter behavior. In contrast, GMF-Mean consistently maintains strong and robust performance at $\lambda = 1.0$ across all task scales, underscoring its robustness. Additional sensitivity analyses—covering different architectures, language models, and vision–language models—are provided in Section C.4.

---

[1]Each task fine-tunes only the CLIP vision encoder. The weights of the classification head are derived from the CLIP text encoder, so altering the classification prompt templates yields different label embeddings and thus different classification heads, enabling task augmentation without additional data.

## 7 CONCLUSION

In this paper, we introduced the Gram-weighted Mahalanobis Fréchet Mean (GMF Mean), a hyperparameter-robust solution for model merging. Building on the observation that the most information in task vectors is concentrated in their principal components, we reformulated model merging as a generalized Fréchet mean problem under task-specific Mahalanobis geometries. This formulation naturally yields a closed-form solution that adapts to the spectral structure of task vectors. Theoretical analysis confirmed its adaptability: GMF Mean faithfully preserves dominant components when tasks are orthogonal and performs singular-value–weighted averages for conflicting directions.

Extensive experiments across vision, language, and vision–language benchmarks demonstrate that GMF Mean achieves competitive performance compared with state-of-the-art model merging approaches. Importantly, our method requires no retraining, no data, and less hyperparameter tuning, making it highly practical for large-scale and resource-constrained scenarios. We view this geometry-aware perspective as a promising direction for model merging, and anticipate that exploring richer metrics, layer- or module-wise geometries, and scalable low-rank or sparse implementations will yield deeper theory and stronger performance.

## 8 ETHICS STATEMENT

This work is conducted solely to advance machine learning research and does not involve any specific ethical concerns.

## 9 REPRODUCIBILITY STATEMENT

We have provided an anonymous code package in the supplementary materials to facilitate the re-generation of the experimental results (see README.md).

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

# A    THEORETICAL ANALYSIS

## A.1    PROOF OF EQ.(9)

To merge multiple task vectors $\{T_1, \ldots, T_K\}$ under the framework of the Gram-weighted Mahalanobis Fréchet mean (GMF Mean), we want to solve the following optimization problem:

$$T^\star \in \arg\min_T \sum_{k=1}^{K} \|T - T_k\|^2_{T_k T_k^\top}, \tag{13}$$

where the generalized Mahalanobis distance is defined with respect to the Gram matrix $T_k T_k^\top$.

This objective admits a closed-form solution as follows:

$$T^\star = \left(\sum_k T_k T_k^\top\right)^\dagger \left(\sum_k T_k T_k^\top T_k\right), \tag{14}$$

where $\dagger$ denotes the Moore-Penrose pseudoinverse.

*Proof.* Expand the objective of GMF Mean:

$$
\begin{aligned}
T^\star \in \arg\min_T & \sum_{k=1}^{K} \|T - T_k\|^2_{T_k T_k^\top} \\
= \arg\min_T & \sum_{k=1}^{K} \operatorname{tr}\left((T - T_k)^\top T_k T_k^\top (T - T_k)\right) \\
= \arg\min_T & \sum_{k=1}^{K} \operatorname{tr}\left(T^\top T_k T_k^\top T - 2T^\top T_k T_k^\top T_k + T_k^\top T_k T_k^\top T_k\right) \\
= \arg\min_T & \operatorname{tr}\left(T^\top \left(\sum_k T_k T_k^\top\right) T - 2T^\top \left(\sum_k T_k T_k^\top T_k\right) + \left(\sum_k T_k^\top T_k T_k^\top T_k\right)\right)
\end{aligned}
\tag{15}
$$

Since each $T_k T_k^\top$ is positive-definite, this defines a convex quadratic optimization problem with the objective function:

$$f(T) = \operatorname{tr}\left(T^\top \left(\sum_k T_k T_k^\top\right) T - 2T^\top \left(\sum_k T_k T_k^\top T_k\right) + \left(\sum_k T_k^\top T_k T_k^\top T_k\right)\right) \tag{16}$$

We obtain:

$$\nabla f(T) = 2\left(\sum_k T_k T_k^\top\right) T - 2\left(\sum_k T_k T_k^\top T_k\right) \tag{17}$$

Using the first-order optimality (and KKT) condition for the unconstrained convex quadratic, we therefore obtain a closed-form solution:

$$
\begin{aligned}
\nabla f(T^\star) &= 0 \\
2\left(\sum_k T_k T_k^\top\right) T^\star &= 2\left(\sum_k T_k T_k^\top T_k\right) \\
\left(\sum_k T_k T_k^\top\right) T^\star &= \left(\sum_k T_k T_k^\top T_k\right) \\
T^\star &= \left(\sum_k T_k T_k^\top\right)^\dagger \left(\sum_k T_k T_k^\top T_k\right).
\end{aligned}
\tag{18}
$$

$\square$

## A.2 PROOF OF EQ.(22)

Consider the singular value decomposition of each task vector $T_k = U_k \Sigma_k V_k^\top$. We focus on a simplified scenario in which only a single task vector $T_i$ exhibits significant energy along a pair of principal directions $(u, v)$, with $u \in U_i$ and $v \in V_i$, while for all $k \neq i$, $u^\top U_k = 0$ and $v^\top V_k = 0$. In this setting, $T_i^\top u = \sigma v$ and $T_i v = \sigma u$ for some singular value $\sigma$, while $T_k^\top u = T_k v = 0$ for $k \neq i$. Under these conditions, it can be shown that the GMF Mean solution $T^\star = \left( \sum_k T_k T_k^\top \right)^\dagger \left( \sum_k T_k T_k^\top T_k \right)$ satisfies:

$$T^{\star \top} u = \sigma v \quad \text{and} \quad T^\star v = \sigma u. \tag{19}$$

*Proof.* We first prove that $T^{\star \top} u = \sigma v$:

$$
\begin{aligned}
T^{\star \top} u &= \left( \left( \sum_k T_k T_k^\top \right)^\dagger \left( \sum_k T_k T_k^\top T_k \right) \right)^\top u \\
&= \left( \sum_k T_k T_k^\top T_k \right)^\top \left( \sum_k T_k T_k^\top \right)^{\dagger \top} u \\
&= \left( \sum_k T_k T_k^\top T_k \right)^\top \left( \sum_k T_k T_k^\top \right)^\dagger u \\
&= \left( \sum_k T_k T_k^\top T_k \right)^\top \frac{1}{\sigma^2} u \\
&= \frac{1}{\sigma^2} \left( \sum_k T_k^\top T_k T_k^\top \right) u \\
&= \frac{1}{\sigma^2} T_i^\top T_i T_i^\top u \\
&= T_i^\top u \\
&= \sigma v.
\end{aligned}
\tag{20}
$$

We can easily prove $T^\star v = \sigma u$ with analogous steps.

$\square$

## A.3 PROOF OF EQ.(21)

Consider the singular value decomposition of each task vector $T_k = U_k \Sigma_k V_k^\top$. In another situation where multiple task vectors share energy along a common pair of singular directions $(u, v)$, the GMF Mean solution $T^\star = \left( \sum_k T_k T_k^\top \right)^\dagger \left( \sum_k T_k T_k^\top T_k \right)$ becomes an adaptive weighted combination:

$$T^{\star \top} u = \underbrace{\left( \sum_k \frac{\sigma_k^2}{\sum_k \sigma_k^2} \sigma_k \right)}_{\text{Weighted Average}} v, \qquad T^\star v = \left( \sum_k \frac{\sigma_k^2}{\sum_k \sigma_k^2} \sigma_k \right) u, \tag{21}$$

where $\sigma_k$ denotes the singular value of $T_k$ in the $(u, v)$ direction.

*Proof.* We first prove that $T^{\star \top} u = \left( \sum_k \frac{\sigma_k^2}{\sum_k \sigma_k^2} \sigma_k \right) v$:

$$T^{\star\top}u = \left(\left(\sum_k T_k T_k^\top\right)^\dagger \left(\sum_k T_k T_k^\top T_k\right)\right)^\top u$$

$$= \left(\sum_k T_k T_k^\top T_k\right)^\top \left(\sum_k T_k T_k^\top\right)^{\dagger\top} u$$

$$= \left(\sum_k T_k T_k^\top T_k\right)^\top \left(\sum_k T_k T_k^\top\right)^\dagger u$$

$$= \left(\sum_k T_k T_k^\top T_k\right)^\top \frac{1}{\sum_k \sigma_k^2} u$$

$$= \frac{1}{\sum_k \sigma_k^2}\left(\sum_k T_k^\top T_k T_k^\top\right) u \qquad (22)$$

$$= \frac{1}{\sum_k \sigma_k^2}\left(\sum_k T_k^\top T_k T_k^\top u\right)$$

$$= \frac{1}{\sum_k \sigma_k^2}\left(\sum_k \sigma_k^2 T_k^\top u\right)$$

$$= \frac{1}{\sum_k \sigma_k^2}\left(\sum_k \sigma_k^3 v\right)$$

$$= \frac{\sum_k \sigma_k^3}{\sum_k \sigma_k^2} v$$

$$= \left(\sum_k \frac{\sigma_k^2}{\sum_k \sigma_k^2}\sigma_k\right) v.$$

We can easily prove $T^\star v = \left(\sum_k \frac{\sigma_k^2}{\sum_k \sigma_k^2}\sigma_k\right) u$ with analogous steps.

$\square$

## A.4 DISCUSS THE ADAPTABILITY OF GMF MEAN VIA GLOBAL PRINCIPAL DIRECTIONS

Define $S$ as the aggregate Gram matrix:

$$S = \sum_{k=1}^K T_k T_k^\top = T_{\text{cat}} T_{\text{cat}}^\top, \qquad T_{\text{cat}} = [T_1, \ldots, T_K]. \qquad (23)$$

Let $S = P\Lambda P^\top$ be its eigendecomposition, with $\Lambda = \text{diag}(\lambda_1 \geq \cdots \geq \lambda_d \geq 0)$ and $P = [p_1, \ldots, p_d]$. Recall the GMF objective from Eq. (8):

$$\mathcal{J}(T) = \sum_{k=1}^K \|(T - T_k)\|^2_{T_k T_k^\top}, \qquad (24)$$

where $\|X\|^2_W := \text{tr}(X^\top W X)$. Let $T^\star$ denote the minimizer of $\mathcal{J}$.

**Curvature along global principal directions.** For any perturbation $\Delta T \in \mathbb{R}^{d\times p}$ around $T^\star$, the second-order increase of $\mathcal{J}$ decomposes along the eigenvectors of $S$:

$$\mathcal{J}(T^\star + \Delta T) - \mathcal{J}(T^\star) = \langle \Delta T, S\,\Delta T\rangle_F = \sum_{i=1}^d \lambda_i \left\|\Delta T^\top p_i\right\|^2_2, \qquad (25)$$

where $\langle A, B \rangle_F := \mathrm{tr}(A^\top B)$. Equivalently, the Hessian with respect to $\mathrm{vec}(T)$ is $H = 2\,(S \otimes I_p)$. Thus, deviations aligned with high-energy global principal components (large $\lambda_i$) incur larger curvature and are penalized more, while low-energy directions have weaker curvature and are naturally shrunk.

**Interpretation.** Equation (23) shows that $S$ is precisely the Gram operator of all tasks stacked together, so its eigenpairs $(p_i, \lambda_i)$ capture the *global* principal directions and their energies. The decomposition in Eq. (25) explains the adaptability of GMF Mean: the optimizer $T^\star$ aligns with directions where tasks collectively concentrate variance, without any manual weighting or hyperparameter tuning.

## A.5 Discuss the Adaptability of GMF Mean via Generalized Singular Value Decomposition

This section provides the analysis about the adaptability of GMF Mean through Generalized Singular Value Decomposition (GSVD) (Paige & Saunders, 1981). Specifically, we consider two task vectors $T_1$ and $T_2$, which can be decomposed as follows:

$$T_1 = U_1 \Sigma_1 V^\top \quad T_2 = U_2 \Sigma_2 V^\top \quad \Sigma_1^\top \Sigma_1 + \Sigma_2^\top \Sigma_2 = I, \tag{26}$$

where $U_1$ and $U_2$ are orthogonal matrices, $\Sigma_1$ and $\Sigma_2$ are diagonal, and $V$ is invertible.

Similarly, the transposed task vectors are given by

$$T_1^\top = V_1 S_1 Y^\top \quad T_2^\top = V_2 S_2 Y^\top \quad S_1^\top S_1 + S_2^\top S_2 = I. \tag{27}$$

Under these conditions, the GMF Mean solution $T^\star = \left(T_1 T_1^\top + T_2 T_2^\top\right)^\dagger \left(T_1 T_1^\top T_1 + T_2 T_2^\top T_2\right)$ can be rewritten as follows along any direction $\hat{v} \in V^{\top^{-1}}$:

$$
\begin{aligned}
T^\star \hat{v} &= \left(T_1 T_1^\top + T_2 T_2^\top\right)^\dagger \left(T_1 T_1^\top T_1 + T_2 T_2^\top T_2\right) \hat{v} \\
T^\star \hat{v} &= \left(Y S_1^\top S_1 Y^\top + Y S_2^\top S_2 Y^\top\right)^\dagger \left(T_1 T_1^\top T_1 \hat{v} + T_2 T_2^\top T_2 \hat{v}\right) \\
T^\star \hat{v} &= \left(Y(S_1^\top S_1 + S_2^\top S_2) Y^\top\right)^\dagger \left(T_1 T_1^\top \sigma_1 u_1 + T_2 T_2^\top \sigma_2 u_2\right) \\
T^\star \hat{v} &= \left(Y Y^\top\right)^\dagger \left(\sigma_1^2 T_1 v + \sigma_2^2 T_2 v\right).
\end{aligned}
\tag{28}
$$

Thus, in the direction of $\hat{v}$, $T^\star$ is a linear combination of the $T_1$ and $T_2$ along $v$. Each task contributes proportionally to $\sigma^2$ within the geometric relationships defined by $\left(Y Y^\top\right)^\dagger$.

## A.6 Empirical Validation for GMF Mean's Adaptability

This section empirically validates the adaptability of GMF Mean through both performance-based visualizations. We first examine the principal angles between Gram matrices of task vectors across network layers. As shown in Figure 3, two clear trends emerge: (1) task vectors become increasingly orthogonal in deeper layers; and (2) the vectors for EuroSAT, SVHN, GTSRB, and MNIST exhibit substantially stronger orthogonality relative to the rest.

We then compare how Task Arithmetic and GMF Mean handle merging conflicts. For each pair of tasks, we designate one as the *Source Task* and the other as the *Target Task*. We merge their task vectors and measure the performance drop of the merged model on the Source Task, relative to the model fine-tuned directly on it. This drop serves as a quantitative indicator of conflict (Sun et al., 2025b).

As summarized in Figure 4, EuroSAT, SVHN, GTSRB, and MNIST consistently exhibit smaller performance degradation than the remaining tasks, which aligns with the theoretical expectation that orthogonality corresponds to low conflict, whereas alignment induces high conflict. Moreover, across both orthogonal and aligned task-vector pairs, GMF Mean consistently incurs lower conflict than Task Arithmetic, demonstrating enhanced robustness and adaptability to diverse directional relationships.

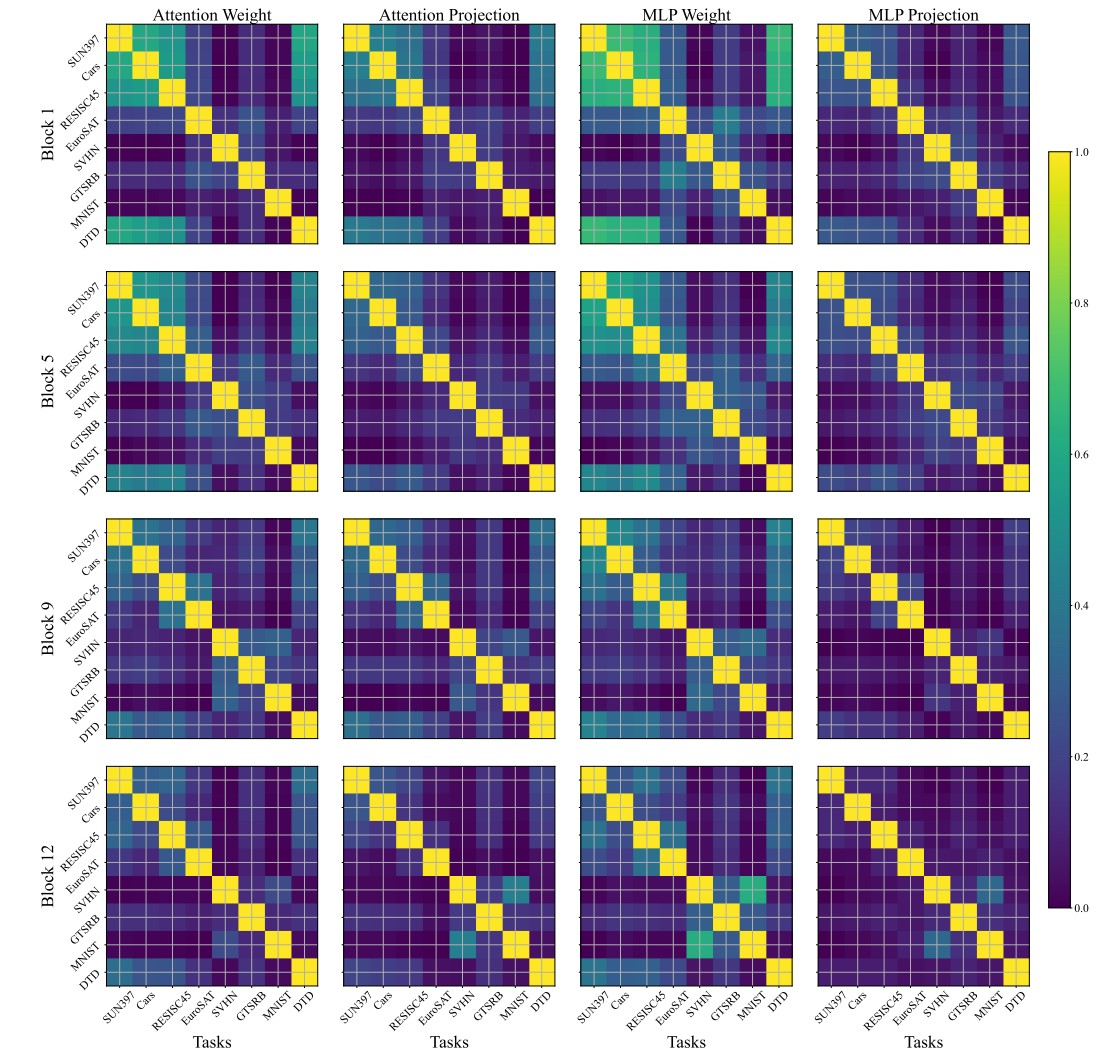

Figure 3: Principal angle between the Gram matrices of two task vectors at various layers for ViT-B/32 models.

Collectively, these empirical findings support the claim that GMF Mean is inherently adaptive to principal-direction conflict and provides more robust and reliable merging behavior than Task Arithmetic.

### A.7 CONDITION TO THE ROBUSTNESS OF HYPERPARAMETER

This section analyzes the hyperparameter robustness of GMF-Mean under assumptions regarding the structure of the Gram matrices. To this end, we examine the Gram matrices of the ViT-B/32 model across different layers to investigate their directional and scaling relationships.

**Exploring the orientations of Gram matrices.** Figure 3 visualizes the principal angles between the Gram matrices of any pair of task vectors. We can observe two clear trends: (1) Task vectors become increasingly orthogonal in deeper layers; and (2) Certain tasks (EuroSAT, SVHN, GTSRB, and MNIST) exhibit significantly stronger orthogonality, while others show higher alignment in specific layers. Empirical evidence shows that this diversity in orientations does not hinder the effectiveness of GMF-Mean, which continues to deliver competitive performance when merging ViT-B/32 models at $\lambda = 1.0$. Moreover, in Section 5 demonstrates that GMF Mean can effectively

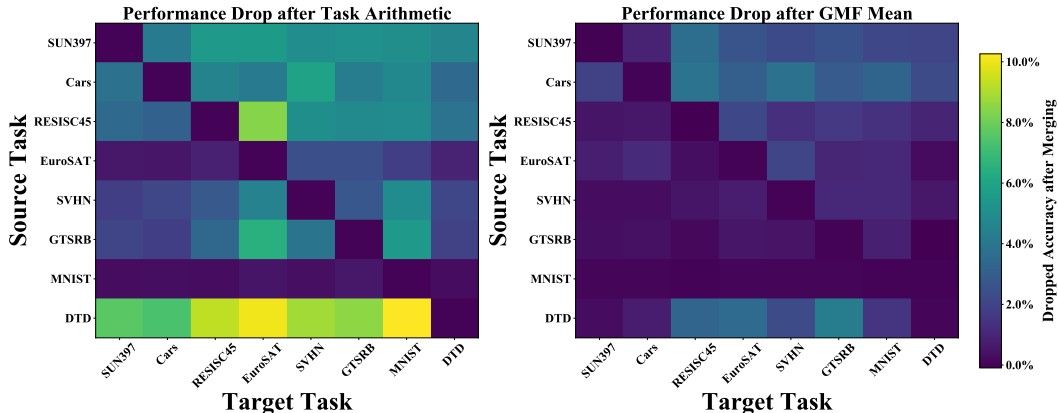

Figure 4: Performance drop on a source task after merging a target task for Task Arithmetic (left) and GMF Mean (right)

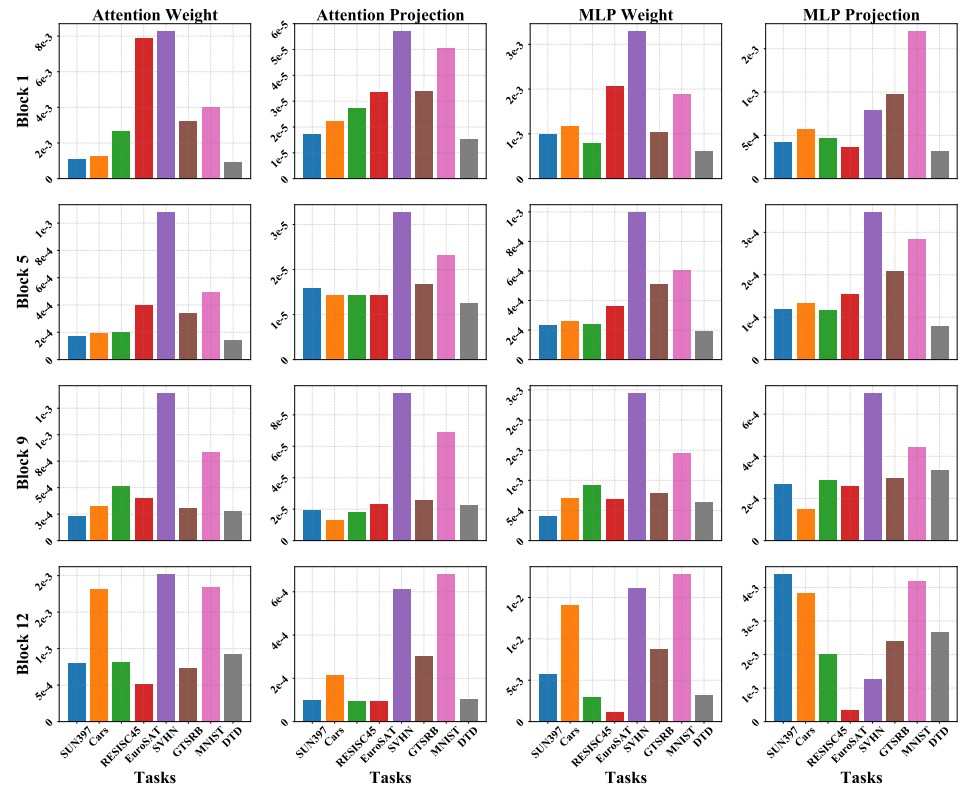

Figure 5: Frobenius norm of Gram matrices for task vectors at various layers for ViT-B/32 models.

handle both orthogonal and aligned Gram matrices. Consequently, we conclude that GMF Mean does not assume any specific alignment or orthogonality of Gram matrices in orientation.

**Exploring the scales of Gram matrices.** Furthermore, we computed the Frobenius norm of different Gram matrices, as shown in Figure 5. The norms exhibit notable variation, with some matrices differing by up to 5 times, yet they remain within the same order of magnitude. Additionally, Figure 6 shows the singular value distributions of the Gram matrices. These distributions follow a typical long-tail pattern, with differences in order of magnitude corresponding to the norm scale. Figure 4 illustrates the correlation between the Normalized Accuracy Improvement (NAI (Marczak et al., 2025)) and the average Frobenius norms across layers. From the results, GMF-Mean does not

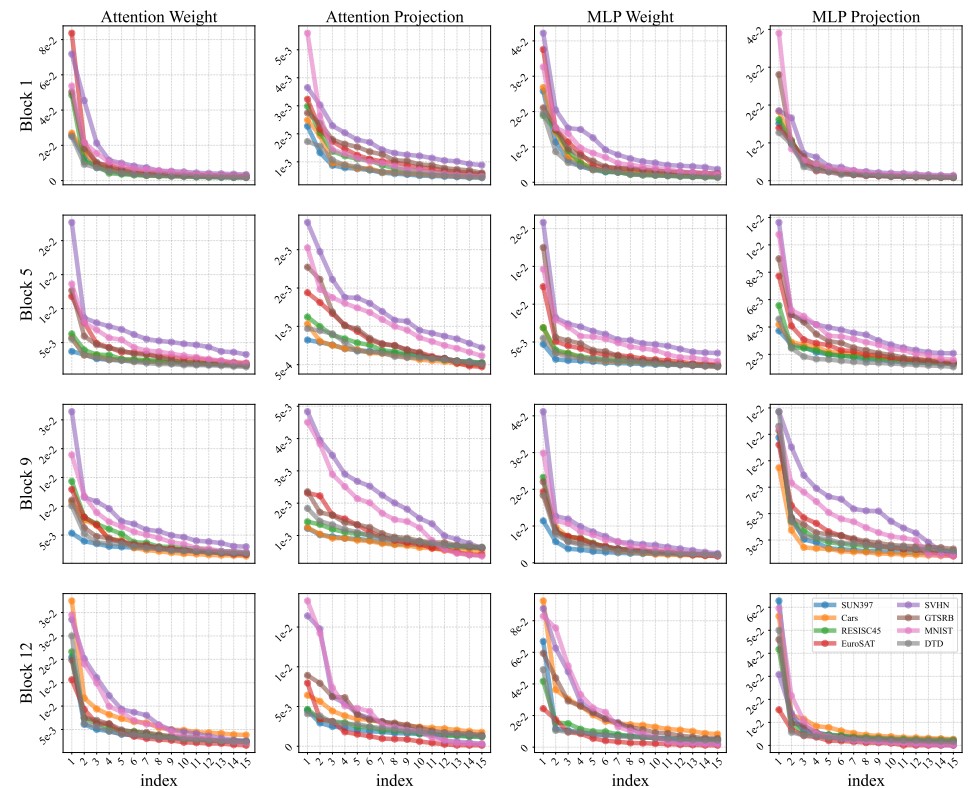

Figure 6: Distribution of singular values of Gram matrices for task vectors at various layers for ViT-B/32 models.

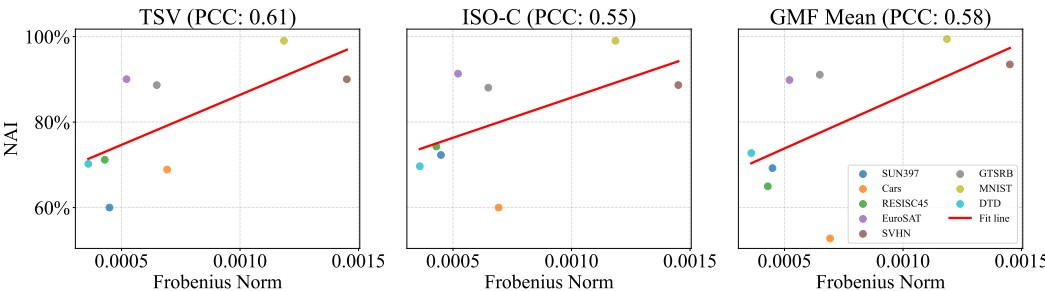

Figure 7: Relationship between NAI scores and the mean Frobenius norms across layers during the merging of eight ViT-B/32 models.

exhibit a stronger bias toward large-norm tasks compared to the existing methods (ISO-C and TSV). Therefore, we can conclude that GMF-Mean is robust to moderate variations in the scale of Gram matrices, as long as their magnitudes remain within a comparable range.

In the context of model merging, since all models are initialized from a well-optimized pretrained network, the loss landscape near $W_{pre}$ is often near-convex (Li et al., 2025), and fine-tuned solutions are typically linearly connected to the pretrained weights (Zhou et al., 2024). Therefore, task vectors do not deviate far and exhibit properties similar to gradients (Sun et al., 2025b). Hence, the assumption that "the magnitude of gram matrices remains within a comparable range" is reasonable and should not be difficult to achieve in model merging.

## B    EXPERIMENT DETAILS

This section provides details of experiments, including the description of the experimental environment, datasets, and baselines.

### B.1    ENVIRONMENT

All experiments detailed in our manuscript and appendix were conducted on a workstation running Ubuntu 16.04, equipped with 18 Intel Xeon 2.60GHz CPUs, 256 GB of memory, and 8 NVIDIA RTX3090 GPUs. Python 3.8 was used to implement all the methods.

### B.2    DATASETS

Our experiments are conducted across vision, language, and vision-language tasks. For vision tasks, we select 8 image classification datasets to evaluate the ability of the merged model, which the details are as follows:

- **SUN397** (Xiao et al., 2016): A scene classification dataset containing 108,754 images across 397 classes. Each class includes at least 100 images.
- **Stanford Cars (Cars)** (Krause et al., 2013): A car classification dataset featuring 16,185 images of 196 car categories. The dataset is evenly split between training and test sets.
- **RESISC45** (Cheng et al., 2017): A remote sensing image classification dataset comprising 31,500 images across 45 scene categories, with approximately 700 images per class.
- **EuroSAT** (Helber et al., 2019): A satellite image classification dataset consisting of 27,000 labeled and geo-referenced images distributed among 10 categories.
- **SVHN** (Netzer et al., 2011): A real-world digit classification dataset derived from house numbers in Google Street View images. It includes 10 classes, with a training set of 73,257 images, a test set of 26,032 images, and an additional 531,131 samples available for extended training.
- **GTSRB** (Stallkamp et al., 2011): A traffic sign classification dataset comprising more than 50,000 images across 43 traffic sign categories.
- **MNIST** (LeCun & Cortes, 2010): A well-known benchmark for handwritten digit classification, containing 60,000 training images and 10,000 test images, evenly distributed among 10 classes of digit numbers.
- **DTD** (Cimpoi et al., 2014): A texture classification dataset consisting of 5,640 images distributed across 47 texture classes, with approximately 120 images per class.

For language, we choose four tasks from the MTEB benchmark (Muennighoff et al.). The information of these datasets is as follows:

- **ArguAna** (Wachsmuth et al., 2018): An argument-retrieval benchmark in which each query is a claim and the task is to retrieve supporting or opposing arguments from a corpus of web-sourced arguments. It evaluates stance-aware semantic retrieval and reranking.
- **HotpotQA** (Yang et al., 2018): A large-scale multi-hop QA dataset built on Wikipedia that requires reasoning across multiple documents and providing supporting sentences. It includes both distractor and full-wiki settings to assess evidence-based reasoning.
- **BIOSSES** (Soğancıoğlu et al., 2017): A biomedical semantic textual similarity dataset consisting of sentence pairs from PubMed abstracts, annotated with human similarity scores on a 0–5 scale. Evaluation typically reports Pearson/Spearman correlations.
- **Banking77** (Casanueva et al., 2020): A fine-grained intent classification dataset in the banking domain, covering 77 customer-service intents over short user queries. It is a standard benchmark for domain-specific intent recognition and sentence classification.

For vision-language tasks, we select the following three caption datasets and three VQA datasets, with the details as follows:

- **COCO Caption** (Chen et al., 2015): A large-scale captioning corpus built on MS COCO, comprising over 330,000 images with five human-written captions per image. It is a standard benchmark for training and evaluating models that generate natural language descriptions from images.

- **Flickr30k Caption** (Plummer et al., 2015): A dataset of 31,000 Flickr images, each paired with five descriptive sentences. It is widely used for both image captioning and bidirectional image–text retrieval.

- **TextCaps** (Sidorov et al., 2020): An OCR-centric captioning benchmark with 145,000 image–caption pairs, where successful captioning requires reading and reasoning over textual content embedded in the scene.

- **OKVQA** (Marino et al., 2019): A knowledge-intensive VQA dataset with over 14,000 questions that require external world knowledge beyond visual perception, designed to assess knowledge-grounded visual reasoning.

- **TextVQA** (Singh et al., 2019): A VQA dataset focused on understanding text within images, containing over 45,000 questions across 28,000 images. Solving it demands OCR and joint visual–textual reasoning.

- **ScienceQA** (Lu et al., 2022): A multimodal multiple-choice QA benchmark with over 21,000 questions spanning biology, chemistry, physics, and related domains, each paired with images and textual explanations to evaluate scientific reasoning.

### B.3 BASELINES.

In our experiments, we compare our methods with several baseline methods. The details of these methods are as follows:

**i) Basic baseline methods:**

- **Pre-trained** directly employs a pre-trained model to predict across multiple tasks. Since it does not incorporate any downstream task-specific information during model training, its performance on downstream tasks is normally suboptimal.

- **Individual**. In this approach, an independent fine-tuned model is used for each task. While it avoids interference between tasks, it cannot perform multiple tasks simultaneously. It serves as a reference *upper bound* for model merging approaches.

- **Traditional MTL** aggregates the original training data from all tasks to train a single multi-task model.

**ii) Test-time training-based methods:**

- **AdaMerging** (Yang et al., 2024b) leverages an unlabeled test set to adaptively learn the merging coefficients at either a layer-wise or task-wise level in Task Arithmetic.

- **AdaMerging++** (Yang et al., 2024b) is an enhanced version of AdaMerging, which integrates the mask of Ties-Merging (Yadav et al., 2023).

- **Surgery** (Yang et al., 2024a) introduces a feature transformation module, trained to align features during the merging process. In this work, we adopt the basic version of Surgery combined with task arithmetic for evaluation.

- **Localize-and-Stitch** (He et al., 2025) optimally combines the strengths of several fine-tuned models by identifying and localizing essential regions within each model before merging.

- **DOGE AM** (Wei et al., 2025) is the DOGE methods applied to AdaMerging (Yang et al., 2024b).

**iii) Training-free methods:**

- **Weight Averaging** directly averages model parameters from multiple tasks into a single model, enabling multi-task learning without additional training.

Table 7: Multi-task performance when merging ViT-B/16 models on eight tasks.

| Method | SUN397 | Cars | RESISC45 | EuroSAT | SVHN | GTSRB | MNIST | DTD | Avg Acc |
|---|---|---|---|---|---|---|---|---|---|
| Pre-trained | 63.8 | 64.6 | 65.7 | 54.5 | 52.0 | 43.3 | 51.7 | 45.1 | 55.0 |
| Individual | 81.8 | 86.8 | 96.9 | 99.7 | 97.8 | 99.1 | 99.7 | 82.0 | 92.9 |
| Weight Averaging | 67.7 | 70.0 | 75.3 | 79.5 | 74.9 | 60.1 | 94.4 | 43.8 | 70.7 |
| Fisher Merging | 68.5 | 69.9 | 75.2 | 80.4 | 73.2 | 61.2 | 94.5 | 50.7 | 71.7 |
| RegMean | 69.1 | 71.6 | 77.6 | 88.8 | 83.7 | 70.2 | 96.9 | 54.6 | 76.6 |
| Task Arithmetic | 61.1 | 65.9 | 74.0 | 76.2 | 88.0 | 73.9 | 98.4 | 53.0 | 73.8 |
| Ties-Merging | 69.1 | 72.5 | 80.5 | 84.0 | 85.0 | 71.5 | 98.1 | 54.9 | 77.0 |
| TATR | 67.4 | 70.4 | 77.9 | 81.7 | 87.6 | 77.2 | 98.3 | 55.6 | 77.0 |
| AWD Merging | 67.8 | 72.7 | 78.7 | 88.5 | 90.9 | 83.6 | 98.9 | 57.1 | 79.8 |
| CAT Merging | 72.9 | 75.9 | 83.1 | 92.8 | 88.2 | 82.7 | 98.8 | 62.7 | 82.1 |
| LOT Merging | 71.0 | 76.2 | 87.6 | 95.8 | 96.5 | 91.9 | 99.2 | 67.0 | 85.7 |
| TSV | 72.8 | 80.4 | 89.1 | 96.6 | 93.9 | 93.9 | 99.3 | 72.7 | 87.3 |
| GMF Mean (ours) | **75.7** | **79.8** | **90.6** | **97.6** | **95.3** | **91.6** | **99.4** | **76.0** | **88.3** |

- **Fisher Merging** (Matena & Raffel, 2022) leverages the Fisher information matrix to assess parameter importance, merging model parameters based on this importance.

- **RegMean** (Jin et al., 2023) refines weight matrices by adjusting and linearly combining rows, utilizing statistical information derived from the training data.

- **Task Arithmetic** (Ilharco et al., 2023) introduces the concept of a "task vector," defined as the difference between fine-tuned model parameters and pre-trained model parameters. Multiple task vectors are then combined and added to the pre-trained model to facilitate multi-task learning.

- **Ties-Merging** (Yadav et al., 2023) eliminates unimportant parameters from the task vector and resolves sign conflicts among parameters, reducing interference during the final task vector merging process.

- **TATR** (Sun et al., 2025b): TATR improves upon Task Arithmetic by restricting the merging of task vectors to a defined trust region, which reduces knowledge conflicts between tasks.

- **Consensus Merging** (Wang et al., 2024a): This method computes a set of masks for each task vector to minimize the distance between the merged model and each fine-tuned model in the parameter space.

- **AWD Merging** (Xiong et al., 2024): AWD Merging generates redundant vectors such that subtracting them from the original task vectors leads to increased orthogonality in the remaining vectors.

- **PCB Merging** (DU et al., 2024) scales task vectors based on the importance of parameters.

- **CAT Merging** (Sun et al., 2025a) alleviates conflicts among task vectors using masking or projection techniques, preserving shared components while suppressing incompatible ones.

- **DOGE TA** (Wei et al., 2025) frames merging as minimizing each expert's loss gap under a shared-subspace constraint, yielding a task-aware, projection-based combination.

- **LOT Merging** (Sun et al., 2025c) enforces alignment between intermediate representations of the merged model and task experts, guiding parameter fusion via feature-level consistency.

- **FR Merging** (Zheng & Wang, 2025) applies frequency-domain filtering to task vectors to suppress noise and stabilize merging, retaining salient, task-relevant components.

### iii) Hyperparameter-robust methods:

- **TSV** (Gargiulo et al., 2025) produces a merged task vector aligned with the principal components of per-task vectors, emphasizing high-variance (informative) directions.

Table 8: Multi-task performance when merging ViT-B/32 models on eight vision tasks with different corruptions.

| Corruption | Method | SUN397 | Cars | RESISC45 | EuroSAT | SVHN | GTSRB | MNIST | DTD | Avg. Acc. |
|---|---|---|---|---|---|---|---|---|---|---|
| Clean (no corruption) | Task Arithmetic | 55.2 | 54.9 | 66.7 | 78.9 | 80.2 | 69.7 | 97.3 | 50.4 | 69.1 |
| | LOT Merging | 67.7 | 67.5 | 85.7 | 94.9 | 93.4 | 89.8 | 98.7 | 63.6 | 82.7 |
| | TSV | 70.1 | 72.1 | 85.9 | 94.3 | 90.9 | 91.2 | 99.2 | 68.8 | 84.1 |
| | GMF Mean (ours) | 71.3 | 69.2 | 83.7 | 94.2 | 93.2 | 92.8 | 99.4 | 69.7 | 84.2 |
| Motion Blur | Task Arithmetic | 23.9 | 19.9 | 32.1 | 41.8 | 80.5 | 60.9 | 97.4 | 22.1 | 47.3 |
| | LOT Merging | 40.6 | 33.6 | 46.6 | 56.5 | 89.6 | 82.6 | 98.8 | 30.6 | 59.9 |
| | TSV | 34.1 | 28.8 | 39.3 | 60.0 | 93.3 | 87.5 | 99.4 | 29.9 | 59.0 |
| | GMF Mean (ours) | 37.0 | 32.8 | 44.6 | 54.9 | 92.2 | 86.6 | 99.3 | 33.1 | 60.1 |
| Impulse Noise | Task Arithmetic | 42.3 | 42.4 | 46.2 | 32.0 | 64.5 | 46.0 | 96.7 | 38.7 | 51.1 |
| | LOT Merging | 57.0 | 52.0 | 58.1 | 41.1 | 75.3 | 55.8 | 98.7 | 48.3 | 60.8 |
| | TSV | 51.3 | 52.5 | 55.7 | 38.6 | 72.7 | 58.0 | 99.0 | 49.2 | 59.6 |
| | GMF Mean (ours) | 50.5 | 50.4 | 48.1 | 37.3 | 77.3 | 57.1 | 99.1 | 48.0 | 58.5 |
| Gaussian Noise | Task Arithmetic | 40.6 | 44.6 | 41.0 | 38.5 | 60.1 | 47.3 | 97.3 | 35.7 | 50.6 |
| | LOT Merging | 55.7 | 55.3 | 54.5 | 43.3 | 69.5 | 60.4 | 98.9 | 43.0 | 60.1 |
| | TSV | 51.2 | 56.3 | 49.6 | 35.8 | 67.0 | 62.2 | 99.2 | 46.4 | 58.5 |
| | GMF Mean (ours) | 53.0 | 54.5 | 42.0 | 38.7 | 75.1 | 61.9 | 99.3 | 46.0 | 58.8 |
| Pixelate | Task Arithmetic | 4.8 | 1.2 | 10.4 | 29.0 | 61.4 | 39.5 | 87.4 | 9.9 | 30.5 |
| | LOT Merging | 8.5 | 2.1 | 18.8 | 32.1 | 69.4 | 53.0 | 94.9 | 10.7 | 36.2 |
| | TSV | 32.0 | 23.7 | 45.4 | 69.0 | 88.1 | 87.2 | 98.9 | 30.2 | 59.3 |
| | GMF Mean (ours) | 31.4 | 21.0 | 44.8 | 70.1 | 91.0 | 88.0 | 99.1 | 30.2 | 59.5 |
| Spatter | Task Arithmetic | 39.7 | 38.7 | 49.6 | 35.6 | 53.6 | 38.5 | 92.0 | 32.2 | 47.5 |
| | LOT Merging | 54.2 | 47.9 | 64.6 | 48.9 | 58.9 | 46.2 | 96.3 | 41.9 | 57.3 |
| | TSV | 66.9 | 69.0 | 84.6 | 94.8 | 84.3 | 81.6 | 99.1 | 62.3 | 80.4 |
| | GMF Mean (ours) | 68.7 | 67.8 | 83.6 | 94.2 | 86.7 | 84.3 | 99.3 | 61.5 | 80.8 |
| Contrast | Task Arithmetic | 30.8 | 25.3 | 20.6 | 25.5 | 52.8 | 27.4 | 85.4 | 32.9 | 37.6 |
| | LOT Merging | 44.5 | 35.0 | 28.4 | 25.1 | 63.8 | 29.7 | 92.9 | 43.5 | 45.4 |
| | TSV | 40.7 | 33.7 | 31.0 | 31.0 | 65.4 | 36.4 | 92.2 | 46.7 | 47.1 |
| | GMF Mean (ours) | 41.8 | 33.9 | 31.8 | 34.3 | 66.7 | 36.1 | 93.8 | 46.3 | 48.1 |
| JPEG Compression | Task Arithmetic | 4.7 | 1.7 | 6.0 | 15.9 | 53.4 | 23.3 | 87.3 | 13.1 | 25.7 |
| | LOT Merging | 5.4 | 1.8 | 6.0 | 13.8 | 60.7 | 25.0 | 93.3 | 14.3 | 27.5 |
| | TSV | 4.6 | 1.7 | 7.0 | 17.9 | 64.6 | 31.8 | 94.0 | 16.4 | 29.8 |
| | GMF Mean (ours) | 4.5 | 1.6 | 6.8 | 23.3 | 66.1 | 32.1 | 95.5 | 16.2 | 30.8 |

# C ADDITIONAL EXPERIMENTS

## C.1 COMPARISON ON ViT-B/16

Table 7 presents the results of various model merging methods using the ViT-B/16 architecture. As we can see, TATR significantly improves the multi-task performance of Task Arithmetic, raising the average performance from 73.8% to 77.0%. Additionally, the zero-shot version also provides a certain degree of improvement, ultimately reaching 74.1%.

## C.2 ROBUSTNESS ANALYSIS OF CORRUPTION

This section explores the robustness of GMF Mean compared to other methods when faced with various types of data corruption. The corruptions considered in this study include Motion Blur, Impulse Noise, Gaussian Noise, Pixelate, Spatter, Contrast, and JPEG Compression. These corruptions simulate real-world distortions that often degrade the performance of vision models. The results in Table 8 reveal that GMF Mean shows robustness under these corruptions. For instance, under Motion Blur, GMF Mean achieves a significant improvement over Task Arithmetic, with a notable accuracy boost from 47.3% to 60.1%. Similarly, for Impulse Noise, GMF Mean surpasses Task Arithmetic, showing a higher average accuracy of 58.5% compared to 51.1%. In the case of Pixelate, GMF Mean performs robustly, achieving 59.5% accuracy, which is competitive with TSV and significantly better than Task Arithmetic, which only reaches 30.5%. This robustness is key to maintaining high performance in real-world scenarios where data may be prone to various distortions.

## C.3 PERFORMANCE VS. NUMBERS OF TASKS

In this section, we follow Consensus Merging (Wang et al., 2024a) to merge ViT models fine-tuned on 14 and 20 tasks. Because the original paper does not provide the exact partition of datasets, we derive task vectors from models we fine-tuned ourselves. Consequently, our results may differ

Table 9: Performance (average accuracy) when merging 14 and 20 vision tasks.

| Method | ViT-B/32 | | ViT-L/14 | |
|---|---|---|---|---|
| | 14tasks | 20tasks | 14tasks | 20tasks |
| Task Arithmetic | 64.3% | 60.2% | 78.9% | 73.6% |
| TATR | 68.3% | 64.3% | 80.1% | 74.5% |
| Consensus TA | 68.5% | 64.6% | 81.3% | 78.2% |
| TSV | 78.8% | 77.1% | 88.6% | 87.9% |
| ISO | 78.6% | 74.2% | 88.7% | 87.0% |
| ISO-CTS | 79.4% | 76.3% | **89.7%** | 88.0% |
| GMF Mean (Ours) | **79.5%** | **77.3%** | 89.3% | **88.4%** |

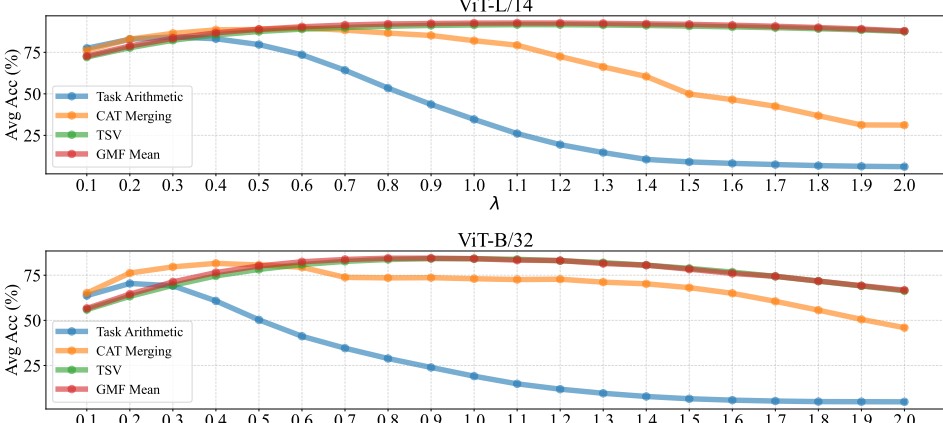

Figure 8: Average accuracy (%) when merging eight ViT-L/14 or ViT-B/32 models versus scaling factor $\lambda$.

slightly from those in the original paper due to variations in dataset partitioning and fine-tuning protocols.

The results in Table 9 show that GMF Mean consistently achieves the best average accuracy across all configurations. On ViT-B/32, GMF Mean achieves 79.5% (14 tasks) and 77.3% (20 tasks), exceeding ISO-CTS by 0.1% and 1.0%. On ViT-L/14, GMF Mean reaches the competitive average performance of 89.3% and 88.4%. These findings highlight the robustness of GMF Mean as the number of merged tasks increases.

### C.4 FURTHER ANALYSIS OF SCALING FACTOR

In Section 6.4, we analyzed the robustness of GMF Mean to the scaling factor by varying $\lambda$ when merging ViT-B/32 models. In this section, we extend the analysis to assess robustness across diverse network architectures, as well as on language and vision–language tasks.

As shown in Figure 8, both GMF Mean and TSV demonstrate strong robustness when merging eight ViT-B/32 and ViT-L/14 models. These results indicate that the inherent hyperparameter robustness of GMF Mean remains stable even under substantial changes in model architecture scale.

Figure 9 (a) shows that performance trends differ considerably across tasks as $\lambda$ increases, reflecting the high heterogeneity and partial conflicts among the tasks. For example, ScienceQA is a multiple-choice question-answering task, while OK-VQA involves open-ended questions requiring single-word answers. Accordingly, as $\lambda$ increases from 0 to 1, performance on OK-VQA gradually declines, whereas performance on ScienceQA steadily improves. Despite these task-specific variations, GMF Mean maintains robust and competitive average performance near $\lambda = 1.0$, as shown in Figure 9 (b). In contrast, the TSV metric achieves optimal performance at $\lambda = 0.8$. These observations further confirm the robustness of GMF Mean to the choice of scaling factor.

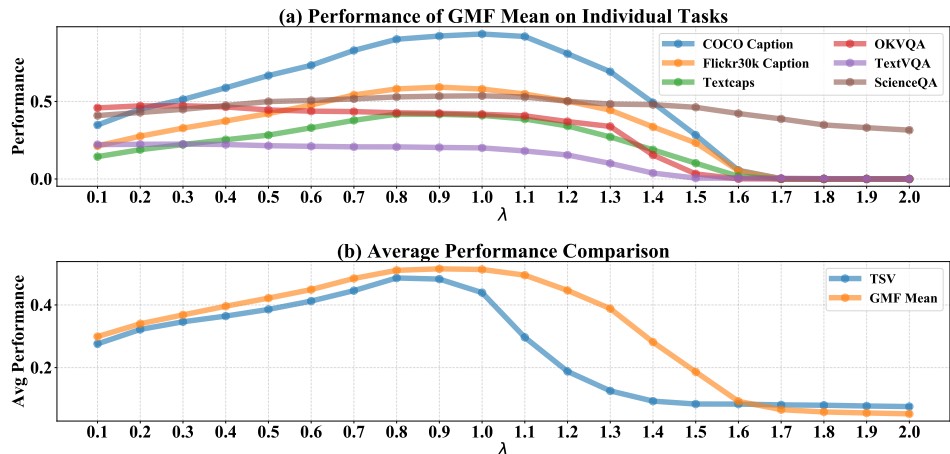

Figure 9: (a) GMF Mean performance on each task when merging BLIP models versus scaling factor $\lambda$. (b) Average performance comparison across all tasks when merging BLIP models versus scaling factor $\lambda$.

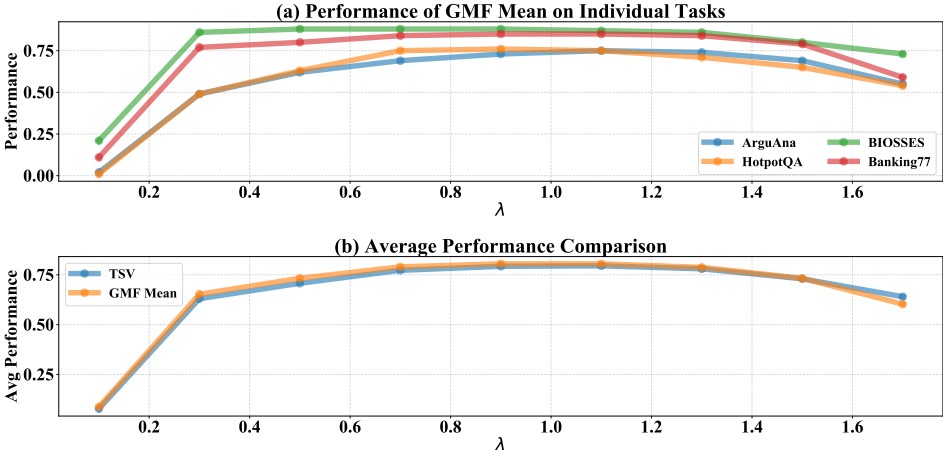

Figure 10: (a) GMF Mean performance on each task when merging Qwen3-8B models versus scaling factor $\lambda$. (b) Average performance comparison across all tasks when merging Qwen3-8B models versus scaling factor $\lambda$.

Figure 10 presents the sensitivity of performance with respect to $\lambda$ when merging Qwen3-8B models. We observe that both GMF-Mean and TSV exhibit strong stability across the entire range of $\lambda$. In particular, all tasks—as well as the overall average—achieve their best performance near $\lambda = 1.0$.

Figure 11 illustrates the sensitivity of performance with respect to $\lambda$ when merging T5-large models. This experiment strictly follows the setup outlined in Ties-Merging (Yadav et al., 2023), where seven T5-large models are merged, each fine-tuned on one of the following NLP tasks: PAWS, QASC, QuaRTz, Story Cloze, WikiQA, Winogrande, and WSC. Since the original checkpoints are not publicly released, we fine-tuned our own T5-large checkpoints using the code framework provided by Yadav et al. (2023). As can be seen, GMF-Mean demonstrates robust performance across all tasks as well as the average performance when $\lambda \geq 1.0$. In contrast, both TSV and Ties-Merging are sensitive to the choice of $\lambda$.

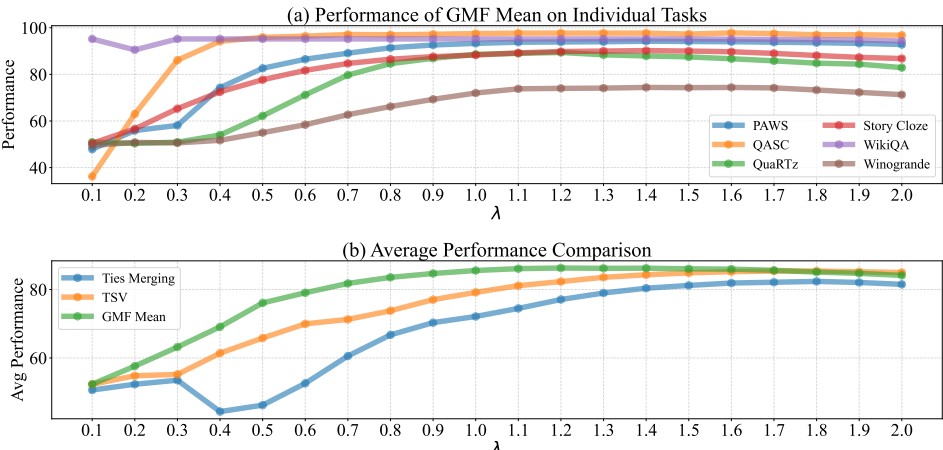

Figure 11: (a) GMF Mean performance on each task when merging T5-large models versus scaling factor $\lambda$. (b) Average performance comparison across all tasks when merging T5-large models versus scaling factor $\lambda$.

## D    LLM USAGE STATEMENT

This work utilized large language models (LLMs) solely for language polishing during manuscript preparation, without involvement in method design or other research issues.

