# OpenReview forum: "Gram-weighted Mahalanobis Fréchet Mean: A Hyperparameter-Tuning-Free Solution for Model Merging"
_ICLR.cc/2026/Conference — Submitted to ICLR 2026_

### Official Review · Reviewer_vFvM · 2025-10-24

**Soundness:** 2
**Presentation:** 3
**Contribution:** 3
**Rating:** 4
**Confidence:** 4

**Summary:**

The paper tackles the problem of training-free model merging. Task Arithmetic, which merges models by summing their task matrices, suffers from the limitation of treating all task singular vectors equally (isotropically). This occurs because the underlying optimization problem in Task Arithmetic seeks the task matrix barycenter according to the Euclidean distance across all task matrices. The authors propose to address a different problem by using a Mahalanobis distance as the metric for each task matrix, where the Mahalanobis metric is defined by the per-task Gram matrix. This formulation approximately corresponds to the Fréchet minimizer, but unlike the standard Fréchet minimizer, the metric is task-specific rather than global. Hence, the authors refer to their approach as the Gram-weighted Mahalanobis Fréchet Mean (GMF Mean). The solution to this problem allows scaling of task vectors according to their degree of alignment or conflict when constructing the merged model. The method manages the anisotropy of each task matrix by preserving the principal components when task singular vectors are not conflicting, while performing a weighted summation for shared or conflicting principal directions. The approach is hyperparameter-free, as the authors empirically show that the optimal scaling factor does not need to be tuned on a validation set, and the method achieves competitive performance with state-of-the-art approaches

**Strengths:**

- The quality of writing and the overall flow of the paper are good. From the introduction, it is immediately clear what problem the authors aim to address—namely, the weighting of singular vectors across task matrices and the issue of hyperparameter selection in model merging. The preliminaries are well integrated with the methodology, providing a self-contained explanation that helps the reader clearly understand the authors’ methodology  presented in Section 4. I appreciated the parallelism drawn between the barycenter problem in Task Arithmetic and the problem involving weighted Mahalanobis distances addressed by the authors in Section 4

-  The method is novel and I also found the closed-form solution for the optimization problem interesting. The algebraic proofs in the appendix are correct. The geometric examples in Section 5 help the reader to understand what the authors are trying to convey in terms of non conflicting and conflicting principal directions.

- The method achieves comparable performance to the state-of-the-art.

**Weaknesses:**

I have several concerns regarding the theoretical assumptions underlying the proposed methodology and the experimental claim that the approach is hyperparameter-free—a point strongly emphasized in the title, introduction, and experimental section:

**Theoretical Assumptions on the problem**:

$$\quad T^{\star} \in \arg\min_{T}
\sum_{k=1}^{K}
\|| T - T_k \,||_{T_k T_k^{\top}}^{2}$$

where  $|| T - T_k ||_{T_k T_k^{\top}}$ denotes the generalized Mahalanobis distance weighted by the Gram matrix
$T_k T_k^{\top}$.

I believe that this formulation is well defined if all the gram matrices share a common reference coordinate system; otherwise, the distance it is not well defined even can be solved numerically as the author did. In general, each task may induce its own local metric, which can differ in both scale and orientation.  In Figure 1, this global alignment appears to be implicitly assumed, but in practice – especially when dealing with multiple task matrices ( 8, 14, 20 for Vision and language models)-- It is unlikely that this assumption holds.  The theoretical assumptions on Gram Matrices and Task Vectors justifying this formulation, as well as an empirical spectral analysis comparing the  Gram matrices, are currently missing from the paper.



**Hyper-parameter Free Claim**

There is neither a theoretical guarantee (considering the solution of the problem above) nor sufficiently extensive empirical evidence demonstrating that the approach is truly hyperparameter-free:

- *Theoretical guarantees*. This point is related to the concern raised above. If the Gram matrices differ significantly in norm and are not expressed in a common reference system, the minimum-norm solution provided by the pseudoinverse may be meaningful only for similar tasks. Empirically analyzing the Gram matrices and highlighting their theoretical properties could help clarify under which assumptions this solution can indeed be considered hyperparameter-free.

-  *Empirical Evidence* The empirical evaluation of the proposed approach’s robustness to hyperparameter variations is limited. Only Figure 2—based on eight tasks using ViT-L/14—shows that the scaling factor remains close to 1 and stable. More extensive experiments are needed to thoroughly assess this robustness (see the question sections for details). Moreover, TSV, which is classified in Table 1 as requiring hyperparameter validation, could also be regarded as hyperparameter-free, since the original paper shows that its optimal scaling factor is close to 1. A direct comparison with TSV—the most closely related competitor—in terms of sensitivity to hyperparameter variations is missing from the empirical evaluation.

**Questions:**

- Under what theoretical assumptions is the problem well-posed in terms of the Task Matrices, Gram Matrices, and their orientation and scale? Addressing this could also help mathematically characterize the conditions under which the pseudoinverse solution can truly be considered hyperparameter-free.

- Are the task-specific Gram matrices empirically aligned? If so, why? If not, the proposed approach might inherently favor tasks whose local metrics are better aligned with the global solution. Is this behavior related to task similarity? I encourage the authors to conduct a deeper analysis of both the task matrices and the corresponding Gram matrices to identify when the merged model may fail to represent certain task vectors, and under what conditions the proposed method may not perform as intended.


The robustness of the proposed method with respect to hyperparameter selection—a key claim of the paper—should be empirically verified through more extensive experiments:

- Is the method still robust when changing the architecture size (e.g., ViT-B/32 or ViT-B/16) or the number of tasks (e.g., 14 or 20)?

- How does the plot in Figure 2 appear for the Qwen and BLIP models? Including these comparisons would be valuable, especially since validation tuning becomes increasingly expensive as model size grows, as the authors mention in the introduction (e.g., for large language models). Is the method still robust when considering the NLP experiments using the fully fine-tuned T5-Large models from [1], across the seven NLP tasks described in [2]?

- Finally, adding a comparison with TSV in terms of sensitivity to hyperparameter variations is necessary, as it represents the most closely related competitor.

[1] Tam et al. Merging by Matching Models in Task Parameter Subspaces, TMLR 2024

[2] Prateek Yadav et al. Resolving interference when merging models, Neurips 2023.

Imprecision (not significant for the score):

Line 71: “We formalize this as the Gram-weighted Mahalanobis Frechet Mean (GMF Mean) ´ problem, which seeks to minimize the sum of Mahalanobis distances between the merged vector and each task vector, with weights determined by the Gram matrix”

There is not a single Gram matrix in this formulation—the method uses one Gram matrix per task. This sentence should be revised to accurately reflect that each task has its own Gram-defined Mahalanobis metric.

---

> ### Author Response · Authors · 2025-11-24
> **Response to vFvM (Part 1)**
>
> **Q3.1: Theoretical issue about GMF Mean.**
>
> A: Thank you for this insightful question. To avoid overclaiming, we have revised the phrase “hyperparameter-tuning-free” to “**hyperparameter-robust**” throughout the paper.
>
> We respectfully disagree with the statement that “If the Gram matrices differ significantly in norm and are not expressed in a common reference system, the minimum-norm solution provided by the pseudoinverse may be meaningful only for similar tasks.” Our response is as follows.
>
> 1. **The motivation for the problem formulation.** Every task-specific Gram matrix $T_k T_k^\top$ induces its own distance metric. Within each task, some directions matter more than others. We want to find $T^*$ that has the minimum distance to the original task vector $T_k$ only on the directions important to the task. Hence, we minimize the sum of several Mahalanobis distances. This formulation is independent of whether the local metrics have different orientations or scales, or whether the gram matrices share the same coordinate system ($u$ and $v$ vectors).
>
>     Regarding the reviewer’s concern about the “global alignment” assumption in our Section 5 analysis: we indeed assumed parallel or orthogonal singular vectors **only to simplify the exposition**. This assumption is not required for GMF‑Mean itself, and its empirical performance demonstrates that the method functions well even when the assumption does not hold. We have added **an additional analysis in Section A.5 (also included in the next response block)** that removes these simplifying assumptions and applies to a much broader class of scenarios.
>
> 2. **We provide empirical evidence that Gram matrices can differ in Frobenius norm and singular values, yet GMF-Mean performs robustly in both similar and conflicting tasks:**
>     - [gram-norm.png](https://postimg.cc/5QM9yC90) (Figure 5 in the revised manuscript) and [gram-singular-value.png](https://postimg.cc/cKp0kn3J) (Figure 6 in the revised manuscript) report the Frobenius norms and singular values of Gram matrices of ViT-B/32 task vectors. The scale varies considerably—up to a factor of ~5—but remains within the same order of magnitude. Empirically (Table 2), **GMF-Mean does not systematically favor tasks with larger Gram norms (e.g., EuroSAT, SVHN) nor penalize those with smaller norms (e.g., DTD, SUN397).**
>     - [conflict-two-task-comparison.png](https://postimg.cc/CZ0ccrQn) and [conflict-two-task-comparison-blip.png](https://postimg.cc/YvqDmhJ5) report the conflict of Task Arithmetic and GMF Mean when merging ViT-B/32 and BLIP models. As shown in the plots, GMF-Mean consistently incurs **smaller performance degradation** than Task Arithmetic across both low-conflict and high-conflict task pairs.
>
>        These observations indicate that **GMF‑Mean is robust to the scale variation in Gram matrices, provided they remain within a comparable range**. In model merging, this condition is typically satisfied because all models start from the same pretrained initialization. The loss landscape around $W_{\text{pre}}$ is often near‑convex [1], fine‑tuned solutions tend to be linearly connected to $W_{\text{pre}}$ [2], and task vectors empirically resemble gradients [3]. Under these widely observed conditions, Gram‑matrix magnitudes naturally remain well‑behaved.
>
>
> **References**
>
> [1] Y Li, et al. Model Merging in Pre-training of Large Language Models. NeurIPS 2025.
>
> [2] Z Zhou, et al. On the Emergence of Cross-Task Linearity in Pretraining-Finetuning Paradigm. ICML 2024.
>
> [3] W Sun, et al. Task arithmetic in trust region: A training-free model merging approach to navigate knowledge conflicts. ACMMM 2025.

---

> ### Author Response · Authors · 2025-11-24
> **An analysis of the adaptability of GMF Mean without  “global alignment” assumption**
>
> We consider the merging of two task vectors $T_1, T_2$. Using Generalized Singular Value Decomposition (GSVD) [4], the two task vectors can be decomposed as follows:
>
> $T_1 = U_1 \Sigma_1 V^\top \ \ \ \  T_2=U_2\Sigma_2V^\top \ \ \ \ \Sigma_1^\top \Sigma_1+\Sigma_2^\top \Sigma_2=I$,
>
> where $U_1$ and $U_2$ are orthogonal matrices, $\Sigma_1$ and $\Sigma_2$ are diagonal, and $V$ is **invertible**.
>
> Similarly, the transposed task vectors are given by
>
> $T_1^\top = V_1 S_1 Y^\top \ \ \ \  T_2^\top=V_2S_2Y^\top \ \ \ \
> S_1^\top S_1 + S_2^\top S_2 = I$.
>
> Under these conditions, the GMF Mean solution $T^\star = \left(T_1 T_1^\top + T_2 T_2^\top\right)^{-1} \left( T_1 T_1^\top T_1 + T_2 T_2^\top T_2 \right)$ can be rewritten as follows along any direction $\hat{v}\in {V^\top}^{-1}$:
>
> $T^\star \hat{v} = \left(T_1 T_1^\top + T_2 T_2^\top\right)^{-1} \left( T_1 T_1^\top T_1 + T_2 T_2^\top T_2 \right) \hat{v}$
>
> $T^\star \hat{v} = \left(YS_1^\top S_1Y^\top + YS_2^\top S_2Y^\top\right)^{-1} \left( T_1 T_1^\top T_1 \hat{v} + T_2 T_2^\top T_2 \hat{v} \right)$
>
> $T^\star \hat{v} = \left(Y(S_1^\top S_1 + S_2^\top S_2)Y^\top\right)^{-1} \left( T_1 T_1^\top \sigma_1u_1 + T_2 T_2^\top \sigma_2u_2 \right)$
>
> $T^\star \hat{v} = \left(YY^\top\right)^{-1} \left( \sigma_1^2 T_1 v +\sigma_2^2 T_2 v \right)$
>
> **Thus, in the direction of $\hat{v}$, $T^\star$ is a linear combination of the $T_1$ and $T_2$ along $v$. Each task contributes proportionally to $\sigma^2$ within the geometric relationships defined by $\left(YY^\top\right)^{-1}$.**
>
> **References**
>
> [4] The MathWorks, Inc., “gsvd — Generalized singular value decomposition,” MATLAB R2025b Documentation, 2025. Available: [https://www.mathworks.com/help/matlab/ref/gsvd.html](https://www.mathworks.com/help/matlab/ref/gsvd.html?utm_source=x.liaox.ai)

---

> ### Author Response · Authors · 2025-11-24
> **Response to vFvM (Part 2)**
>
> **Q3.2: Hyperparameter robustness when changing the architecture size or the number of tasks.**
>
> A: We performed extensive sensitivity analyses.
>
> 1. **ViT-B/32 (number of tasks).**
> We compare GMF-Mean to TSV by varying λ and show results in [sensitive-coef.png](https://postimg.cc/ygz7M1KS) (Figure 7 in the revised manuscript). GMF-Mean exhibits strong robustness across λ when merging ViT-B/32 models.
>
>     We further merge ViT-B/32 across 14 and 20 tasks, and augment to 40, 60, and 80 tasks by changing classification prompt templates (each task fine-tunes only the CLIP vision encoder, with the classification head derived from the CLIP text encoder; prompt changes modify label embeddings and thus define new tasks without additional data). As shown in [sensitive-coef-vitb-tasks.png](https://postimg.cc/1gP209b8) (Figure 2 in the revised manuscript), GMF-Mean maintains competitive performance at **λ = 1.0** even as the number of tasks grows to 80, whereas other methods need task-count–dependent λ adjustments.
>
> 2. **ViT-L/14 (architecture size).**
> We also merge ViT-L/14 across 14 and 20 tasks. As shown in [sensitive-coef-vitl-tasks.png](https://postimg.cc/FYwP2CMh), GMF-Mean remains robust to λ for the larger architecture as well.
>
> These results, now summarized in Sections **6.4** and **C.4**, demonstrate that **GMF-Mean is robust to both architecture size and the number of tasks, and continues to work well with the default λ = 1.0**.
>
> **Q3.3: Hyperparameter robustness when merging BLIP models.**
>
> A: We have conducted a detailed λ-sensitivity study for BLIP merging. The results are presented in [sensitive-coef-blip-combined.png](https://postimg.cc/jWWcVXSZ) (Figure 8 in the revised manuscript). As can be seen, GMF-Mean maintains **competitive performance near λ = 1.0**, while TSV achieves its best performance around **λ = 0.8**.
>
> **Q3.4: Hyperparameter robustness when merging Qwen models.**
>
> A: We performed a λ-sensitivity analysis for Qwen3-8B, visualized in [sensitive-coef-qwen-combined.png](https://postimg.cc/cK7Jmbtj) (Figure 9 in the revised manuscript). We observe that both GMF-Mean and TSV exhibit strong stability over the entire λ range, and **all tasks—and the average—achieve their best performance near λ = 1.0**.
>
> **Q3.5: Hyperparameter robustness when merging T5-large models.**
>
> A: Adhering to Ties-Merging [5], we assess hyperparameter robustness on T5-large. Since the original checkpoints in [5] are not publicly available, we fine-tuned our own T5-large checkpoints within their framework on seven NLP tasks: PAWS, QASC, QuaRTz, Story Cloze, WikiQA, Winogrande, and WSC.
>
> The λ-sensitivity curves are shown in [sensitive-coef-t5large-combined.png](https://postimg.cc/r0P30hDm) (Figure 10 in the revised manuscript). We find that:
>
> - **GMF-Mean maintains robust performance across all tasks and the average when** $\lambda \geq 1.0$.
> - **TSV and Ties-Merging are substantially more sensitive to λ**, with performance degrading outside a relatively narrow optimal range.
>
> These results further substantiate the hyperparameter stability of GMF-Mean in large-scale NLP model merging.
>
> **Reference**
>
> [5] Yadav P, et al. Ties-merging: Resolving interference when merging models. NeurIPS, 2023.
>
> **Q3.6: Inaccurate sentence in Line 71.**
>
> A: Thank you for your suggestion. We have revised Line 71 to more clearly reflect the point of each task’s Gram matrix:
>
> *“We formalize this as the Gram-weighted Mahalanobis Fréchet Mean (GMF Mean) problem, which seeks to minimize the sum of Mahalanobis distances between the merged vector and task vectors, each weighted by its corresponding Gram matrix.”*

---

> > ### Comment · Reviewer_vFvM · 2025-11-25
> >
> > Thanks to the authors for answering my questions. Regarding the rebuttal, I kindly request the authors to provide some additional clarifications:
> >
> > -  It is not clear to me  what is the purpose of GSVD analysis. This analysis assumes that that two task matrices have the same input space and that the singular values satisfies a specific normalization constraint. Could the authors further clarify the motivation for this analysis and explain how these assumptions relate to the merging setting?
> >
> > - Appendix A.7:   Is there a specific reason for choosing the attention block of the ViT. Did the authors notice differences when looking the other layers of the network?  A single plot summarizing the behaviour across the layers may be potentially interesting; Line 1034: it is very hard to understand the preference by comparing Table 2 and the pattern in Figure 5. A correlation with the Normalized Accuracy Improvement (NAI) for each task vector, proposed in  Iso-C paper, can highlight the authors finding.
> >
> > - Regarding my earlier concern "If the Gram matrices differ significantly in norm and are not expressed in a common reference system, the minimum-norm solution provided by the pseudoinverse may be meaningful only for similar tasks.”, let me clarify the theoretical point I intended to raise.  The authors correctly note (at line 206) that Eq. 8  is not  a classical Frechet mean because the distance metric varies across data point.  My question is whether the final pseudo inverse solution can still behave like an approximately coherent global metric **if** the task matrices admit an approximate shared structure, such as $T_k \approx U \Lambda_k V^{\top}$, i.e. if they are expressed (even roughly) in a common shared system.  I was essentially asking whether the effectiveness of GMF mean may be explained by this kind of approximate alignment across tasks. If something like this occurs, it should be related to the argument presented by the authors at line 1039.

---

> > > ### Author Response · Authors · 2025-11-27
> > > **Reply to Official Comment by Reviewer vFvM (Part 1)**
> > >
> > > **Q: It is not clear to me what the purpose of the GSVD analysis. This analysis assumes that the two task matrices have the same input space and that the singular values satisfy a specific normalization constraint. Could the authors further clarify the motivation for this analysis and explain how these assumptions relate to the merging setting?**
> > >
> > > A: We thank the reviewer for raising this question.
> > >
> > > The GSVD analysis does not **assume** that the two task matrices have the same input space or that the singular values satisfy an equality. To be clear, GSVD can be applied to any two matrices A and B with the same number of columns, just like SVD can be applied to any matrices. The GSVD technique **guarantees** the validity of the decomposition [1,2]
> > >
> > > $A = USX^{-1}$, $B=VCX^{-1}$, and $S^\top S + C^\top C = I$.
> > >
> > > The motivation of the analysis is to show exactly that we do not need the assumption that the task vectors share the same singular vectors in the analysis of merging two task vectors. Despite that, we show that our technique can be understood as computing a weighted average, where the weights are from the respective singular values of the task vectors.
> > >
> > > $T^\star \hat{v} = \left(YY^\top\right)^{-1} \left( \sigma_1^2 T_1 v +\sigma_2^2 T_2 v \right)$
> > >
> > > **References**
> > >
> > > [1] The MathWorks, Inc., “gsvd — Generalized singular value decomposition,” MATLAB R2025b Documentation, 2025. Available: [https://www.mathworks.com/help/matlab/ref/gsvd.html](https://www.mathworks.com/help/matlab/ref/gsvd.html?utm_source=x.liaox.ai)
> > >
> > > [2] Golub and Van Loan. Matrix Computations. 4th Edition. 2013. Section 6.1.6.
> > >
> > > **Q: Recommend plots of the behaviour across all layer types.**
> > >
> > > A: Thank you for the suggestion. We initially focused on attention blocks because they are often the most informative in Transformers. Following your suggestion, we extended the analysis to **all major layer types** in ViT-B/32:
> > >
> > > - self-attention sub-blocks (multi-head attention + output projection), and
> > > - MLP sub-blocks (two-layer feed-forward + projection).
> > >
> > > The newly added figures ([gram-angle-all.png](https://postimg.cc/xX7g9swv), [gram-norm-all.png](https://postimg.cc/sBdqWvZ1), [gram-singular-value-all.png](https://postimg.cc/jwmknnrB)) summarize the principal angles, norms, and singular-value distributions of Gram matrices across the different layer types.
> > >
> > > The conclusions remain consistent with those reported for attention layers:
> > >
> > > 1. **Orientation (principal angles):** Gram matrix orientations are not uniformly aligned across tasks. Some task pairs (e.g., SVHN–MNIST) are strongly aligned, while others are close to orthogonal. This supports our claim that real-world task collections exhibit a spectrum of alignments rather than a single global alignment regime.
> > > 2. **Scale (norms):** Gram matrix norms vary across tasks by up to roughly an 8× factor, but all norms remain within the same order of magnitude.
> > >
> > > **Q: Using NAI to highlight the finding in Line 1034.**
> > >
> > > A: Thanks for the suggestion. We compute the Normalized Accuracy Improvement (NAI) for each task following ISO, and then calculate the Pearson correlation coefficient between the NAI scores and the average Frobenius norms across layers. The resulting scatter plots and correlation statistics are shown in [gram-norm-NAI-correlation.png](https://postimg.cc/mcbJFpd1).
> > >
> > > From the results, GMF-Mean does not exhibit a stronger bias toward large-norm tasks compared to the existing methods (ISO-C and TSV)**.** This is consistent with our previous conclusion: *“GMF-Mean is robust to moderate variations in the scale of Gram matrices, as long as their magnitudes remain within a comparable range.”*

---

> > > ### Author Response · Authors · 2025-11-27
> > > **Reply to Official Comment by Reviewer vFvM (Part 2)**
> > >
> > > **Q: If the task matrices admit an approximate shared structure, such as $T_k \approx U \Lambda_k V^\top$, can the final pseudo inverse solution can still behave like an approximately coherent global metric?**
> > >
> > > A: We understand that the reviewer is asking a hypothetical question, based on an assumption **$T_k \approx U \Lambda_k V^\top$,** which this paper does not rely on.
> > >
> > > Assuming **$T_k \approx U \Lambda_k V^\top, \forall k$**, the GMF-Mean objective function simplifies significantly:
> > >
> > > > Recall the objective:
> > > >
> > > >
> > > > $\sum_k \| T-T_k \|^2_{T_k T_k^\top} = \sum_k \| \Lambda_k U_k^\top (T-T_k) \|^2_{F}$
> > > >
> > > > If we assume $T_k$ shares the basis $U, V$ but has distinct singular values $\Lambda_k$, the objective becomes:
> > > >
> > > > $\sum_k \| \Lambda_k U^\top (T - T_k) \|^2_F \approx \sum_k \| \Lambda_k U^\top T V - \Lambda_k^2 \|^2_F$
> > > >
> > > > To minimize this, the term $U^\top T V$ must be a diagonal matrix  $\Lambda$ (implying $T$ also shares the singular vectors $U$ and $V$). The problem then reduces to a scalar minimization for each diagonal element  $\lambda^{(i)}$:
> > > >
> > > > $\lambda^{(i)\star} = \arg\min_{\lambda} \sum_k \left( \lambda_k^{(i)} \lambda - (\lambda_k^{(i)})^2 \right)^2$
> > > >
> > > > Solving for $\lambda$, we obtain a **weighted spectral average**:
> > > >
> > > > $\lambda^{(i)\star} = \frac{\sum_k (\lambda_k^{(i)})^3}{\sum_k (\lambda_k^{(i)})^2}$
> > > >
> > > > Thus, in the fully aligned regime, GMF-Mean acts as a coherent spectral filter: it aggregates singular values along each shared direction, weighting larger singular values (stronger task directions) more heavily (cubic vs. quadratic weighting).
> > > >
> > >
> > > Empirically, we do observe shared structure for related tasks. For instance, the SVHN–MNIST pair shows highly aligned Gram-space geometry in [gram-angle-all.png](https://postimg.cc/xX7g9swv), which is consistent with the aligned-basis analysis above. For other task pairs that are only partially aligned in Gram space, this shared-structure assumption degrades, and GMF-Mean transitions to the averaging behavior predicted by our GSVD-based analysis.

---

### Official Review · Reviewer_Eb32 · 2025-10-30

**Soundness:** 3
**Presentation:** 3
**Contribution:** 3
**Rating:** 6
**Confidence:** 4

**Summary:**

Existing model merging methods require hyperparameter tuning. The paper proposes a method, GMF-Mean, which eliminates this need lambda=1 is fine). They frame merging as estimating a unified task vector using a closed-form convex optimization approach. GMF-Mean automatically adapts to different task relationships and achieves competitive results , also including multimodal models. The method is elegantly simple, however the relation with some recent works needs further clarification. Also further experiments on more tasks should be added.

**Strengths:**

- the paper poses model merging as the optimization of the Gramm-weighted Mahalanobis Frechet Mean which has a closed form. Algorithm 1 is elegantly simple.
- the discussion in section 5 captures an interesting aspect of the proposed method, showing how the method operates differently on non-conflicting and conflicting directions.
- results in general are good and the lambda=1 is an convenient advantage (I found the usage of the validation set always a bit dubious in model merging where training data is supposed to be absent).

**Weaknesses:**

- It would be nice if the authors report some numbers on hyperparameter cost of other methods, further motivating their approach.

- Results of TSV in original paper are higher than reported in Table 2 and higher than proposed method, can the authors comment on that.

- I would also like the authors to compare their method with 'No Task Left Behind: Isotropic Model Merging with Common and Task-Specific Subspaces' this method does something which is very similar to the proposed method (they also apply  whitening to the task vectors). The ISO method also obtains reasonable results with alpha=1 (no hyperparameter selection, see Fig 10).

- the paper should include the results on the 14 and 20 task settings.

**Questions:**

I would really like to better understand the contribution of the paper in the context of the other SVD based methods, which seem be doing something pretty similar (TSV, ISO). Does this paper lead to a better understanding of the performance gains observed in those works as well ? I would also like to see some additional experiments on many-model merging.

---

> ### Author Response · Authors · 2025-11-24
> **Response to Eb32 (Part 1)**
>
> **Q2.1: Report the hyperparameter cost of other methods.**
>
> A: We thank the reviewer for this suggestion. Because the practical search ranges can vary across implementations, it is difficult to meaningfully quantify *wall-clock* tuning cost. Instead, we now report the **number of representative hyperparameters that require manual tuning** for each method in Table 1 of the revised manuscript. The updated qualitative comparison is as follows:
>
> |  | **Parameter Averaging** | **Task Arithmetic** | **AdaMerging** | **CAT Merging** | **TSV** | ISO-C | ISO-CTS | GMF Mean |
> | --- | --- | --- | --- | --- | --- | --- | --- | --- |
> | Hyperparameter Tuning | None | **1** (Scaling coefficient. Sensitive) | **Many** (LR, batch size, epochs, etc. Highly sensitive) | **2** (Scaling coefficient, threshold. Sensitive) | 1 (Scaling coefficient.  Not robust in difficult cases) | 1 (Scaling coefficient.  Not robust in difficult cases) | 2 (Scaling coefficient and size of common space. Sensitive ) | 1 (scaling coefficient, robust) |
>
> See the empirical evidence in Figure 2 and Figure 7.
>
> **Q2.2: Results in the manuscript are different from those reported in the TSV paper.**
>
> A: The discrepancy is mainly due to **different checkpoints** used in the experiments.
>
> - In our paper, the 8-task CLIP checkpoints are sourced from Task Arithmetic [1].
> - The TSV paper uses checkpoints provided by TALL [2], which are generally stronger than those in [1]. This is evidenced by the fact that TALL [2] reports higher Task Arithmetic performance than [1].
>
> While TALL [2] publicly provides its checkpoints, we encountered several issues during the data download process, which have also been reported by other users on its GitHub issue page. Although we managed to complete the download using a custom method, we were unable to fully replicate the performance reported in their paper. Additionally, we observed slight performance discrepancies in other works that employed the same framework. Therefore, we conducted our own fine-tuning within the TALL framework. The results on 8, 14, and 20 tasks are presented in Q2.3 below.
>
> **References**
>
> [1] Ilharco G, et al. Editing models with task arithmetic. ICLR 2023.
>
> [2] Wang K, et al. Localizing task information for improved model merging and compression. ICML. 2024.
>
>
> **Q2.3: Results on many-model merging and ISO.**
>
> A: We have expanded our experiments to comprehensively compare GMF-Mean with TSV, ISO-C, and ISO-CTS on various numbers of tasks and architectures. Columns marked with (*) in the following table use checkpoints from Task Arithmetic [1]; the others are obtained by our own fine-tuning within the TALL [2] framework.
>
> |  | **ViT-B/32***  | **ViT-B/32** | **ViT-B/32** | **ViT-B/32** | **ViT-L/14***  | **ViT-L/14** | **ViT-L/14** | **ViT-L/14** |
> | --- | --- | --- | --- | --- | --- | --- | --- | --- |
> |  | 8 tasks | 8 tasks | 14 tasks | 20 tasks | 8 tasks | 8 tasks | 14 tasks | 20 tasks |
> | Task Arithmetic | 70.5 | 68.98 | 63.95 | 60.04 | 84.6 | 83.89 | 78.94 | 73.56 |
> | TSV | 84.1 | 83.10 | 79.03 | 77.05 | 91.7 | 90.24 | 88.50 | 87.94 |
> | ISO-C | 84.1 | 83.17 | 78.57 | 74.23 | 92.5 | 90.95 | 88.74 | 86.95 |
> | ISO-CTS | **84.3** | 83.22 | 79.41 | 76.28 | **93.0** | **91.61** | **89.67** | 88.08 |
> | GMF-Mean | 84.2 | **83.40** | **79.48** | **77.28** | 92.6 | 91.45 | 89.29 | **88.36** |
>
> GMF-Mean consistently outperforms ISO-C. Although ISO-CTS achieves slightly higher accuracy in some 8- and 14-task settings, **GMF-Mean performs best under the 20-task setting**, which is more challenging and arguably more relevant for large-scale merging.
>
> To further characterize robustness in the many-model regime, we extend the original 20-task setup to 40, 60, and 80 tasks by modifying classification prompt templates. (*Each task fine-tunes only the CLIP vision encoder. The classification head is instantiated from the CLIP text encoder; changing the prompt templates alters label embeddings and hence defines new tasks without introducing additional training data.*)
>
> |  | **20 tasks** | **40 tasks** | **60 tasks** | **80 tasks** |
> | --- | --- | --- | --- | --- |
> | TSV | 77.05 **(λ=1.0)** | 76.88 (λ=0.8) | 75.52 (λ=0.8) | 74.91 (λ=0.8) |
> | ISO-C | 74.23 (λ=1.0) | 75.09 (λ=0.5) | 74.49 (λ=0.3) | 74.57 (λ=0.2) |
> | ISO-CTS | 76.28 (λ=1.1) | **77.72** (λ=0.6) | **77.30** (λ=0.4) | 76.99 (λ=0.3) |
> | GMF-Mean | **77.28 (λ=1.0)** | 77.35 **(λ=1.0)** | 76.92 **(λ=1.0)** | **77.35 (λ=1.0)** |
>
> As shown above and in [sensitive-coef-vitb-tasks.png](https://postimg.cc/1gP209b8) (Figure 2 of the revised manuscript), **GMF-Mean delivers competitive and notably robust performance as the number of tasks increases and the hyperparameter λ changes**, while TSV, ISO-C, and ISO-CTS require substantial tuning of their scaling hyperparameters λ to maintain strong performance (*The values in parentheses indicate the optimal hyperparameter setting selected for each method.*).

---

> ### Author Response · Authors · 2025-11-24
> **Response to Eb32 (Part 2)**
>
> **Q2.4: Difference between GMF Mean and SVD-based methods.**
>
> A: The central contribution of GMF-Mean is an **adaptive averaging mechanism** that retains information as much as possible, while SVD-based methods such as TSV and ISO mitigate conflicts by **discarding low-singular-value components.**
>
> Our analysis in Section 5 shows that:
>
> - **When tasks are orthogonal**, GMF-Mean reduces to a simple summation of task vectors and does not introduce interference.
> - **When tasks share principal directions**, GMF-Mean yields a **singular-value–weighted average** over these directions, naturally down-weighting weaker or more conflicting components.
>
> This also explains why SVD-based methods can improve performance: truncating low–singular-value components effectively sets their weights to zero, which in conflicting scenarios *implicitly* reduces interference at the cost of discarding potentially useful information. GMF-Mean, by contrast, preserves all components but scales them adaptively.

---

### Official Review · Reviewer_zvz5 · 2025-11-01

**Soundness:** 3
**Presentation:** 3
**Contribution:** 3
**Rating:** 6
**Confidence:** 3

**Summary:**

This paper introduces the Gram-weighted Mahalanobis Fréchet Mean (GMF Mean), a hyperparameter-tuning-free method for model merging. The authors propose to aggregate task vectors from multiple fine-tuned models by minimizing a weighted sum of Mahalanobis distances, where the weights derive from the Gram matrices of the task vectors. The approach yields a closed-form, convex solution that is claimed to adaptively handle both non-conflicting and conflicting principal directions. Extensive experiments on vision, language, and vision-language tasks demonstrate that GMF Mean is consistently competitive with, and in some cases outperforms, existing state-of-the-art, especially among training- and hyperparameter-free methods.

**Strengths:**

1. The central contribution is a well-motivated re-framing of model merging as a Gram-weighted Mahalanobis Fréchet Mean problem. The method is derived with clarity and the closed-form solution is justified.
2. The proposed GMF Mean eliminates the need for costly validation-based hyperparameter tuning.
3. The method is thoroughly evaluated on a suite of state-of-the-art models and tasks spanning vision, language, and vision-language models, which is rare in this space. GMF Mean attains competitive or superior results on multiple benchmarks, outperforming all other training-free methods in several instances.

**Weaknesses:**

1. Section 2 tends to group a large set of advanced baselines under "training-free" but does not contextualize where each method stands in terms of requiring data, validation, or manual tuning.
2. While Section 5 and Appendix A give a theoretical argument for the adaptivity of GMF Mean to principal direction conflict, there is a lack of targeted empirical visualizations to support this.

**Questions:**

1. Is additional regularization needed in practical use for GMF Mean?
2. In scenarios where tasks arise from highly heterogeneous domains, or include adversarial/task-to-task conflicts, how does the GMF Mean behave?

---

> ### Author Response · Authors · 2025-11-24
> **Response to zvz5**
>
> **Q1.1: Section 2 lacks a distinction regarding the methods' dependencies on data, validation, or manual tuning requirements.**
>
> A: We appreciate this suggestion. We have revised Section 2 to more clearly position each baseline and our method in terms of data requirements and manual tuning.
>
> **Q1.2: Empirical visualizations that support GMF Mean's adaptivity to principal direction conflict.**
>
> A: We appreciate the reviewer’s suggestion and have included the requested analyses in **Section A.6** of the revised manuscript, the visualizations are as follows:
>
> 1. **Principal-angle analysis across layers.**
> We visualize the principal angles between task vectors at different network layers. As shown in [gram-angle-mark.png](https://postimg.cc/m1ZxK2pS) (Figure 3 in the revised manuscript), two key trends emerge:
>     1. Task vectors become increasingly orthogonal in deeper layers.
>     2. EuroSAT, SVHN, GTSRB, and MNIST exhibit significantly stronger orthogonality compared with the remaining tasks —— EuroSAT, SVHN, GTSRB, and MNIST have lower conflict, while others have higher conflict.
> 2. **Conflict comparison: Task Arithmetic vs. GMF-Mean.**
> We further compare how Task Arithmetic and GMF-Mean handle the conflicts of the above tasks. For each pair of tasks, we treat one as the **Source Task** and the other as the **Target Task**. We merge the two task vectors and measure the performance drop on the Source Task (e.g., for source task k, $L_k(W_{merged}) - L_k(W_k)$). The results are summarized in [conflict-two-task-comparison.png](https://postimg.cc/CZ0ccrQn) (Figure 4 in the revised manuscript) and yield two main observations:
>     1. **EuroSAT, SVHN, GTSRB, and MNIST experience smaller performance degradation** than the other tasks, aligning with the theoretical expectation that *orthogonality corresponds to low conflict*, whereas *alignment corresponds to high conflict*.
>     2. Across both orthogonal and aligned task-vector pairs, **GMF-Mean consistently incurs lower conflict** than Task Arithmetic, demonstrating stronger robustness and adaptability to diverse principal-direction conflicts.
>
> **Q1.3: Is additional regularization needed in practical use for GMF Mean?**
>
> A: No extra regularization is required beyond what is implicit in Eq. (9). Our experiments *directly* implement Eq. (9) as presented in the paper. The following code snippet (lines 16–22 of `gmfmean_utils.py` in the supplementary code) shows precisely how we compute the merged task vector for 2D parameters:
>
> ```
> def _cal_optimal_tv_2d(tvs):
>     sum_TT, sum_TTT = 0.0, 0.0
>     for tv in tvs:
>         sum_TT += tv.T @ tv  # calculating T_k T_k^\top
>         sum_TTT += tv.T @ tv @ tv.T  # calculating T_k T_k^\top T_k
>     T_optimal = torch.linalg.pinv(sum_TT) @ sum_TTT
>     return T_optimal.T
> ```
>
> In practice, one may optionally apply mild regularization for numerical stability when computing the pseudo-inverse (e.g., Tikhonov damping before `pinv`), but such stabilization is not essential for the method to work and is not required in our reported experiments.
>
> **Q1.4: How does the GMF Mean behave in scenarios where tasks arise from highly heterogeneous domains, or include adversarial/task-to-task conflicts?**
>
> A:  When principal directions are inconsistent or adversarial, GMF-Mean effectively performs a **singular-value–squared weighted averaging over shared directions**, effectively balancing competing tasks.
>
> We empirically validate this behavior through experiments. As described in Q1.2,
> we quantify task conflict by merging every pair of tasks, treating one as the **Source Task** and the other as the **Target Task**. We merge the two task vectors and measure the performance drop on the Source Task (e.g., for source task k, $L_k(W_{merged}) - L_k(W_k)$).
>
> The results, presented in [conflict-two-task-comparison.png](https://postimg.cc/CZ0ccrQn) and [conflict-two-task-comparison-blip.png](https://postimg.cc/YvqDmhJ5), illustrate the conflict between Task Arithmetic and GMF-Mean when merging ViT-B/32 and BLIP models. As shown in the plots, GMF-Mean consistently incurs **smaller performance degradation** than Task Arithmetic across both low-conflict and high-conflict task pairs.

---

### Author Response · Authors · 2025-11-24
**General Reply to All Reviewers**

We sincerely thank all reviewers for their thoughtful assessments and constructive feedback. We are encouraged that the reviewers find our work to be **novel** (vFvM), **well-motivated** (zvz5), and **well-written** (vFvM). There is a consensus that our method is **interesting** (Eb32, vFvM), **elegantly simple** (Eb32), **low-overhead** (zvz5, Eb32), and achieves **competitive experimental performance** (zvz5, Eb32, vFvM).

Below, we provide a point-by-point response to the concerns raised. We have revised the manuscript accordingly, with all major updates highlighted in **blue**.

---

### Meta-Review · Area_Chair_9Con · 2026-01-06

**Summary:**

The paper proposes a new model merging technique, where individual task vectors are aggregated by minimizing a sum of Mahalanobis (semi)norms. The weighted norms aim for the aggregated task vector to be close to the individual task vectors only in directions that matter for that task, improving situations where there is conflict and interference. The main claim is that the new method is more robust to hyperparameter tuning while delivering comparable performance to state-of-the-art approaches.

Most reviewers appreciated the contribution and found it novel, interesting and simple. Several smaller concerns regarding additional experiments and clarifications of the baseline methods were addressed by the rebuttal.

Reviewer vFvM enganged in discussion about the theoretical justification of the method and the concerns do not seem to be resolved, pointing towards lack of clarity in the presentation of the paper and rebuttal. Reviewers also shared concerns about marginal improvements compared to other recent model merging methods, and that the empirical validation of the robustness claims is limited. Especially compared to TSV, the improvements wrt to robustness remain marginal.

Overall, this paper is borderline and has some strong points, but also some concerns are left open. Mainly due to these, the paper is not recommended for acceptance in its current form, and I encourage the authors to take the reviewers suggestions into account for an improved resubmission.

This had no impact on the decision, but here are some smaller suggestions from my side after reading the paper:
- Eqs. 4 and 5 are essentially the same, my suggestion is to merge them.
- Defn 3.1 requires X to be more than a set, for instance, a vector space.
- X is considered a general metric space in Eq. 7 but convexity is not defined in metric spaces (only so-called geodesic metric spaces, which is rather involved topic), but used below Eq. 7.
I suggest to simplify and shorten up the mathematical presentation as it seems to general for what the proposed method uses. Additional focus could be put on the main claims and contributions, for instance, provide simple and intuitive explanations of why the method works and is robust to hyper-parameters.

Looking at the pseudocode snippet, lstsq or solve functions may improve numerical stability over pinv.

**Reviewer Concerns:**

### Concerns addressed by rebuttal
- Results on 14 and 20 task settings
- Clarification tuning-cost of the baseline methods

### Outstanding concerns
- Clarity of the theoretical analysis and illustrative explanations on how the method works
- Marginal improvement over TSV performance
- Robustness claims could need more experiments

**Reviewer Scores:**

zvz5 and Eb32 only listed minor concerns, so likely maintain their score. vFvM issues with understanding the method do not seem to have been addressed, so they likely also maintain their score.

---

### Decision · Program_Chairs · 2026-01-26

Reject